# Inhibitory circuit motifs in *Drosophila* larvae generate motor program diversity and variability

Jacob Francis[1]☯, Caius R. Gibeily[1]☯, William V. Smith🄳[1]☯*, Isabel S. Petropoulos[1], Michael Anderson[1], William J. Heitler[1], Astrid A. Prinz[2], Stefan R. Pulver🄳[1,3]*

**1** School of Psychology and Neuroscience, University of St Andrews, St Andrews, United Kingdom, **2** Department of Biology, Emory University, Atlanta, Georgia, United States of America, **3** Institute for Behavioural and Neural Sciences, Centre of Biophotonics, and Centre for Biological Diversity, University of St Andrews, St Andrews, United Kingdom

☯ These authors contributed equally.
* wvs@st-andrews.ac.uk (WVS), sp96@st-andrews.ac.uk (SRP)

## Abstract

How do neural networks generate and regulate diversity and variability in motor outputs with finite cellular components? Here we examine this problem by exploring the role that inhibitory neuron motifs play in generating mixtures of motor programs in the segmentally organised *Drosophila* larval locomotor system. We developed a computational model that is constrained by experimental calcium imaging data. The model comprises single-compartment cells with a single voltage-gated calcium current, which are interconnected by graded excitatory and inhibitory synapses. Local excitatory and inhibitory neurons form conditional oscillators in each hemisegment. Surrounding architecture reflects key aspects of inter- and intrasegmental connectivity motifs identified in the literature. The model generates metachronal waves of activity that recapitulate key features of fictive forwards and backwards locomotion, as well as bilaterally asymmetric activity in anterior regions that represents fictive head sweeps. The statistics of inputs to competing command-like motifs, coupled with inhibitory motifs that detect activity across multiple segments generate network states that promote diversity in motor outputs, while at the same time preventing maladaptive overlap in motor programs. Overall, the model generates testable predictions for connectomics and physiological studies while providing a platform for uncovering how inhibitory circuit motifs underpin generation of diversity and variability in motor systems.

## Introduction

Neural networks controlling locomotion must balance the need to generate diversity in motor outputs with the need to constrain outputs at critical moments. How locomotor systems balance this need using a finite number of shared components is a long-standing question in motor systems research. Gaining insight into how rhythmic

**Data availability statement:** All calcium imaging-related data from Regions of interest (.csv), electrophysiology-related files (.csv), NeuroSIM model files (.nrsm), and modelling-related data files (.csv) are available from our PURE database (doi: 10.17630/779141ce-c26a-483b-bfee-4f12cf71d7b2). Raw fluorescence multi-image TIFF files from longer time scale experiments are available upon request. All inquiries about data availability can be submitted to the Research Data and Innovation Services team at University of St Andrews (research-data@st-andrews.ac.uk).

**Funding:** This work was supported by 1) a Biotechnology and Biological Sciences Research Council (BBSRC) EASTBIO CASE PhD studentship awarded to J.F. (BB/M010996/1) (https://biology.ed.ac.uk/eastbio) 2) a University of St Andrews Rector's Fund Award to C.R.G. (https://tinyurl.com/Rectors-Fund), 3) a BBSRC EASTBIO CASE PhD studentship awarded to W.V.S. (BB/M010996/1)(https://biology.ed.ac.uk/eastbio) and 4) a collaborative Research Grant funded by the Global Office at University of St Andrews and The Halle Institute for Global Research at Emory University awarded jointly to A.A.P. and S.R.P. https://tinyurl.com/StA-Emory-CRG. The funders had no role in study design, data collection and analysis, decision to publish, or preparation of the manuscript.

**Competing interests:** We have read the journal's policy and the authors of this manuscript have the following competing interests: W.J.H created and sells both Neurosim and Dataview.

motor systems solve this problem provides a knowledge base for understanding conserved principles of decision-making across different nervous systems.

Most motor circuits contain neurons that participate in the control of multiple motor programs, and other neurons that are dedicated to a single program. Mechanisms enabling motor program selection among competing options in such circuits have been identified in several model organisms, including competitive disinhibition in larval *Drosophila* [1], reciprocal inhibition in molluscan feeding circuits [2] and vertebrate spinal networks [3], and descending pathways [4,5]. Behavioural studies have also revealed that nested central pattern generators (CPGs) can be organised hierarchically and interact to orchestrate sequences of behaviours [6]. Inhibition between circuit components controlling different motor programs has been observed in many circuits [7,8]. However, this motif has also been contrasted with observations of circuits in which most components are thought to participate in multiple programs, and selection between different motor outputs occurs at the level of circuit neuromodulation [9]. Although some key insights have been informed by research in several systems, our understanding of the circuit mechanisms underlying the generation and regulation of diversity in rhythmic motor programs remains fragmented, without clear consensus about conserved principles across animal phyla.

The *Drosophila* larval locomotor system represents an ideal system to study how motor program diversity is generated and regulated. The larval central nervous system (CNS) is relatively small (12–15,000 neurons) and community-led connectomics efforts have generated complete wiring diagrams for large areas of the CNS, including the entire brain [10–13]. Despite its relatively small size, the larval CNS generates an impressive diversity of qualitatively different motor programs using shared neuromuscular components. Intact animals generate bilaterally symmetric waves of muscle contractions enabling forwards and backwards locomotion, as well as bilaterally asymmetric head-sweeping and bending motor programs for turning and rolling (reviewed in [14]). Since these animals feel their way across and through changing terrain, they must be able to quickly interrupt ongoing activity and flexibly adjust motor programs on a cycle-by-cycle basis. The isolated larval CNS spontaneously generates fictive versions of nearly all motor programs observed in intact animals, and the underlying motor patterns can be comprehensively imaged using genetically encoded calcium indicators (GECIs) [15–17]. In isolation, the CNS spontaneously switches between motor programs, each with a characteristic pattern of motor activity. The frequency and duration of these motor patterns varies across and within individuals [17]. The functional roles of identified cells and circuit motifs underlying particular motor programs have been characterised using a combination of connectomics, electrophysiology, imaging, and behaviour analysis (reviewed in [18]). Spontaneous generation of a diverse array of motor programs, which in turn show variability in duration and frequency, appears to lie at the core of central pattern generation in *Drosophila* larvae [17]; however, relatively little is known about the central mechanisms that underlie rhythm generation and how those mechanisms are coordinated to regulate how and when different types of larval motor programs are spontaneously generated.

Detailed, cell-by-cell, experimental analysis of circuit function is one approach to uncovering mechanisms generating diverse sets of motor outputs. Another complementary approach is to create computational models that synthesise and abstract features from experimental data. These models can then generate insight into circuit function, while providing testable predictions for experimentalists. Previous studies have developed dynamical systems-based models of larval behaviour [19,20] and Wilson-Cowan-type coupled oscillators that generate wave-like activity [21]. These abstract models have provided key insights into how behaviours and fictive behaviours can arise in *Drosophila* larvae and isolated nervous systems, but they are constrained largely by behavioural outputs as opposed to anatomical and physiological data. More recently, models that incorporate comprehensive, detailed connectomic data have been developed and used to explore how anatomical architectures can be tuned to recapitulate patterned activity within single segments of the larval abdominal nerve cord [12]. In parallel, Hodgkin-Huxley type conductance-based models of identified neurons have been developed that recapitulate intrinsic excitability profiles [22,23] and anatomical features [24]. These more detailed computational approaches have also been fruitful, but they have yet to reveal fundamental aspects of the larval locomotor system, such as plausible origins of rhythm generation and circuit architectures for segregating motor programs. Computational models of the larval locomotor system that are constrained by physiological and anatomical data to an extent, while also synthesising and combining the best aspects of all these approaches, would provide platforms for exploring how motor program diversity and variability could be generated. Models in this 'Goldilocks' conceptual zone (not too abstract, not too detailed) can then also provide testable predictions for further experimental work.

Here we first present an analysis of diversity and variability in fictive locomotory patterns expressed by the isolated *Drosophila* larval CNS, using optical measurements of CPG activity from GECIs. We then develop a computational model of the *Drosophila* locomotor system as a way to explore how diversity and variability in CPG output can be generated and regulated through inhibitory circuit motifs. The model was designed to represent populations of neurons by single-compartment model neurons that incorporate Hodgkin-Huxley type voltage-gated calcium channels and are connected with physiologically plausible excitatory and inhibitory synapses. The model is constrained by published work and by our own analysis of new experimental work. We examine how different types of biologically plausible inhibitory neurons can be used to generate oscillations, track activity patterns, and segregate competing motor programs. Simulated optogenetic manipulations of inhibition within the model are compared to similar manipulations in experimental preparations. Overall, the model recapitulates several key features of the experimental preparation, gives insight into the potential role of inhibitory motifs in the system, and provides testable predictions for further experimental work.

## Results

### Analysis of activity patterns in isolated CNS preparations

The isolated *Drosophila* CNS spontaneously generates rhythmic activity within abdominal and thoracic regions; the activity of motor neurons innervating muscles in each hemisegment can be measured by imaging the activity of glutamatergic neurons in regions of interest within each hemisegment using GECIs (Fig 1Ai-iv) [17]. Previous work has characterised activity patterns using GCaMP3 [25]. Here, we examined activity patterns in preparations expressing GCaMP6 in glutamatergic neurons using OK371-GAL4. Isolated CNS preparations expressing GCaMP6 exhibited a diversity of fictive motor programs similar to those seen in previous work using other GCaMP variants, including fictive forwards and backwards metachronal waves as well as bilaterally asymmetric activity in thoracic and anterior abdominal regions (Fig 1B). Fictive forwards and backwards locomotion are distinguished by bilaterally symmetric calcium waves that progress from the posterior abdomen to the thorax and vice versa, respectively (Fig 1C, note dark green and black dots above traces). Fictive head sweeps are expressed as bilaterally asymmetric activity in thoracic and anterior abdominal regions (Fig 1C, light green and blue dots above traces). In addition to forwards waves, backwards waves, and bilaterally asymmetric activity, isolated CNS preparations also produced isolated bursts of activity in anterior and posterior segments (magenta and orange dots, respectively). See Methods for specific criteria used to classify each activity pattern.

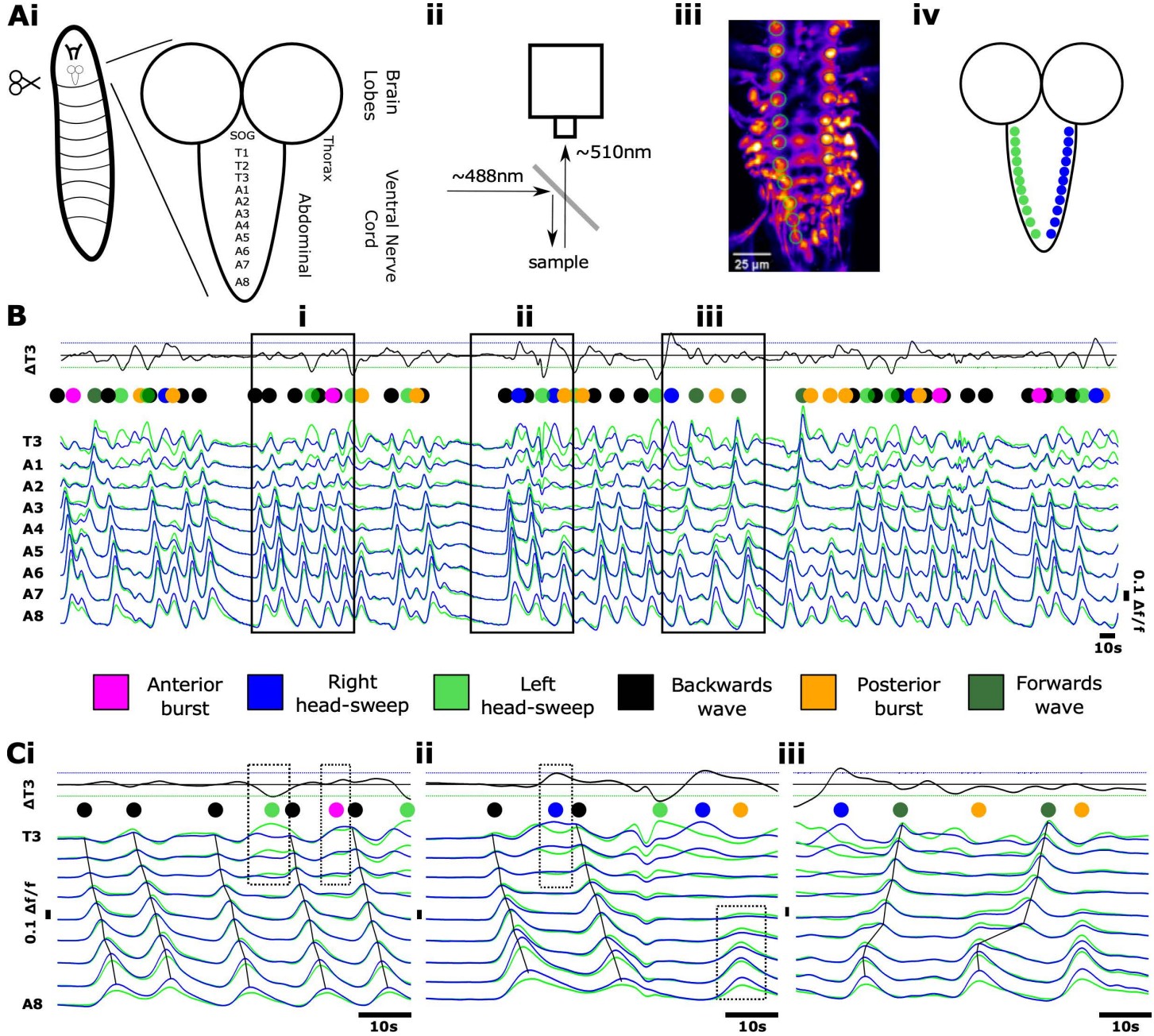

**Fig 1. Imaging fictive locomotion in isolated central nervous system preparations.** (A) Graphical summary of experimental methods to image neuronal activity using genetically encoded Ca²⁺ indicators in the isolated central nervous system (CNS) of 3rd instar larvae including dissection procedure and schematic of CNS **(i)**, fluorescence imaging set up **(ii)**, sample image with ROIs (circles) drawn on motor neuron neuritic regions **(iii)**, and colour schematic for traces **(iv)**. (B) Overview of fictive locomotor activity. Each trace represents the average pixel intensity within an ROI (Aiv) with a baseline correction (Δf/f). Data from ROIs on left (green) and right (blue) sides of the CNS are overlaid and shown for ROIs on segments A8 to T3. An additional trace "ΔT3" was calculated from T3 right subtracted from T3 left as a measure of bilateral asymmetries in thoracic regions. (**Ci-Ciii**) Expanded views of activity patterns shown in **(B)**. Coloured dots denote activity patterns: fictive forwards waves (dark green), fictive backwards waves (black), left and right fictive head-sweeps (blue and light green, respectively) and anterior and posterior bursts (magenta and orange, respectively). Dashed boxes in (i) and (ii) indicate examples of left and right head-sweeps and anterior and posterior bursts (see Methods for criteria used). 10.17630/779141ce-c26a-483b-bfee-4f12cf71d7b2.

First, we examined the extent of temporal overlap in CNS activity patterns. Forwards and backwards waves were, in large part, mutually exclusive and segregated in time. Activity in medial segments (i.e., abdominal segments A1-A6) never showed evidence of simultaneous forwards and backwards waves or of more than one wave progressing through the network (0% of 339 forwards waves and 0% of 382 backwards waves, n = 15 animals). However, in the posterior segments A7 and A8/9 we did observe some instances of partial overlap in which the end of a backwards wave coincided in time with the start of a forwards wave (mean ± S.E.M = 4.3 ± 1.5% of 721 total wave-like events, n = 15 animals). The converse, namely overlap in anterior segments as a forwards wave ended and a backwards wave started, was not observed.

Fictive head sweeps overlapped in time with initiation of backwards waves in A1 in approximately half of all backwards wave events (mean ± S.E.M = 49.7 ± 7.9% of 382 backwards waves, n = 15 animals). For instance, the fourth backwards wave in Fig 1Ci starts with a fictive left head sweep (green dot followed by black dot above the trace), and the second backwards wave in Fig 1Cii starts with a fictive right head sweep (blue dot followed by black dot). In Fig 1Cii, there were also head sweeps that did not initiate backwards waves (blue and green dots). There was high inter-preparation variability in this type of overlap, with some preparations showing head-sweep-backwards wave overlap in only 10.3% of events, while others showed overlap in 100% of backwards waves. We conclude that forwards and backwards wave motor programs are predominantly mutually exclusive but that slight temporal overlaps between forwards and backwards waves can spontaneously occur in posterior regions. We also conclude that initiation of head-sweeps often (but not always) overlaps in time with initiation of backwards waves in anterior regions.

When examining the dynamics of forwards and backwards waves, we noted substantial variability in the duration of waves and in the ways in which waves were initiated. Variability was present both within and across preparations (Fig 2). Forward waves appeared to propagate through medial and anterior segments after a short burst of largely synchronous activity in segments A6-A8 (Fig 2A, 2E). Within preparations, the dynamics of forwards wave propagation varied (Fig 2C). Backwards waves propagated through medial and posterior segments after bilaterally symmetric activity within hemisegments or after bilaterally asymmetric activity (fictive head sweep) in thoracic regions (Fig 2B, 2F). As with forwards waves, propagation dynamics also varied within preparations (Fig 2D).

As noted in previous publications, CNS preparations also produced bursts of activity in posterior and anterior segments (Fig 1 [17,26]. The timing, amplitude and duration of these events qualitatively resembled the initiation of forwards waves (posterior bursts) and backwards waves (anterior bursts), but these events did not subsequently trigger propagation of waves through the entire network. Posterior and anterior bursts sometimes occurred at moments when the network transitioned from one type of motor program to another (mean ± S.E.M = 32.6 ± 8.5% of 247 posterior and anterior bursts, n = 15 animals), with the remainder occurring within a bout of the same program or before periods of quiescence. Regardless of when these events occurred in relation to ongoing motor programs, when either posterior or anterior bursts were occurring, no wave-like activity was present in other parts of the network (mean ± S.E.M = 100.0 ± 0.0%, n = 15 animals).

Overall, these analyses provide several constraints for computational modelling of the larval CNS. A realistic model should be able to: 1) spontaneously generate all observed motor programs with similar dynamics across the A-P axis, 2) spontaneously transition between these programs, 3) show temporal overlap between head-sweeps and backwards waves in some, but not all instances of backwards waves, and 4) show evidence of mutual exclusion of forwards and backwards wave programs. However, perhaps the most striking feature of isolated CNS preparations is that there is both temporal diversity and variability in the frequency and duration of motor programs within and across preparations [17], and a computational model should therefore reflect this diversity.

To characterise these aspects for use as modelling constraints, we imaged CPG activity in multiple intact isolated CNS preparations for extended time periods (30 min), then measured the type and frequency of the motor program produced (see Methods for genotypes and criteria used to define activity patterns). We first measured the instantaneous frequency of each observed motor pattern within 15 preparations (Fig 3A), defined as the inverse of the period since the preceding occurrence of the same pattern. Since the system switches among motor programs so frequently, these data do not

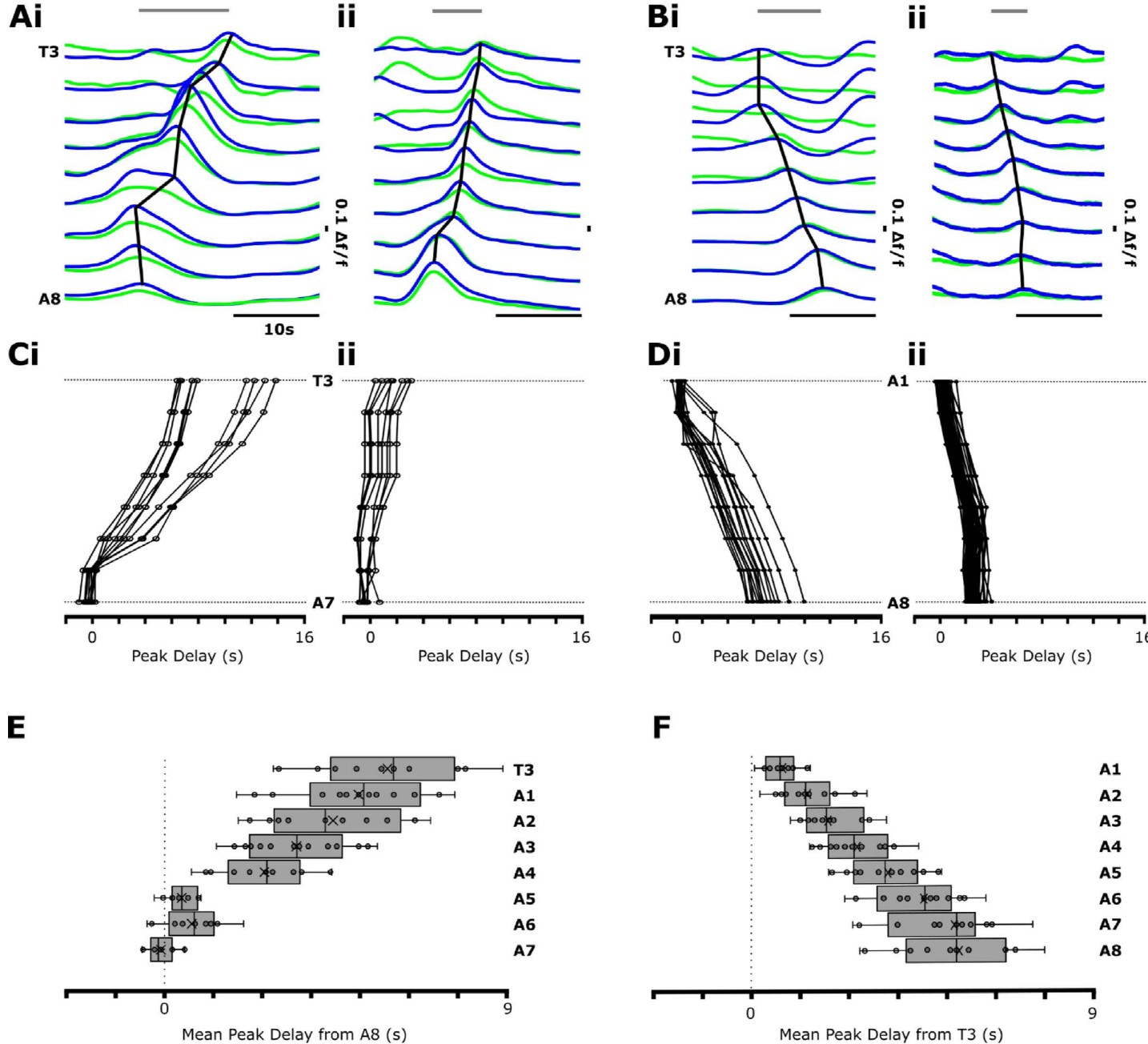

**Fig 2. How fictive forwards and backwards waves propagate varies within and across individuals and across body segments. (A)** Fictive forwards waves show variability in wave propagation. Slow (**i**) and fast (**ii**) example waves are shown. Grey line at top indicates wave duration. **(B)** Fictive backwards waves also show variability with slow (**i**) and fast (**ii**) segmental propagation. **(C)** Data from two example preparations showing intra-animal variability in propagation in slow (**i**) and fast (**ii**) fictive forwards waves. **(D)** Same as in C, but for backwards waves in two experiments. **(E)** Mean peak segmental (T3-A7) delay from A8 of fictive forwards waves across all preparations showing near synchronous posterior segmental peaks. Box-plot shows median (line) and mean (x). **(F)** Mean peak segmental (A1-A8) delay from T3 of fictive backwards waves across all preparations. N = 15 preparations in (E) and **(F)**. 10.17630/779141ce-c26a-483b-bfee-4f12cf71d7b2.

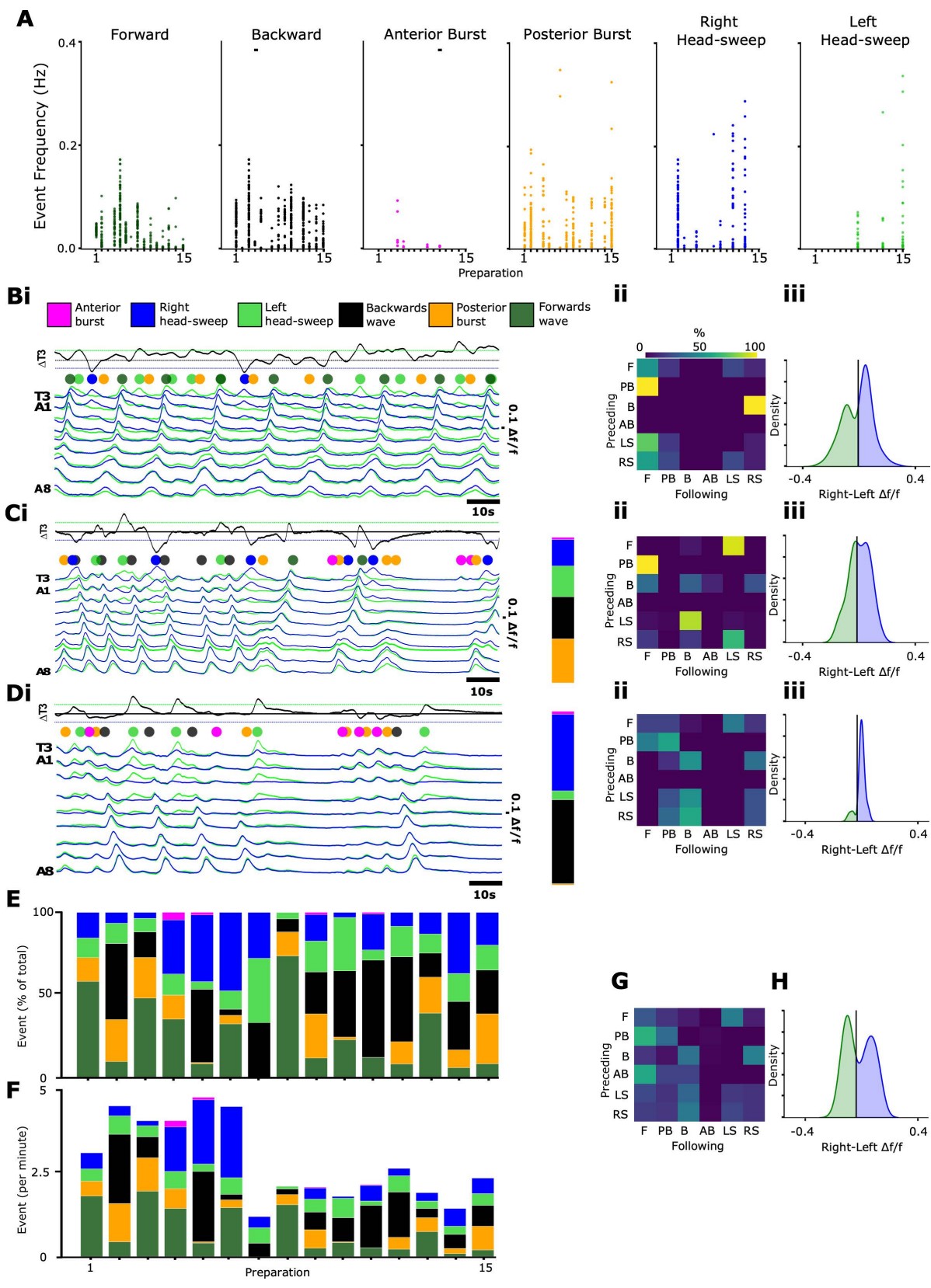

**Fig 3. Quantitation of diversity and variability in motor output in isolated CNS preparations. (A)** Instantaneous frequency of each activity pattern (defined as inverse of time since preceding occurrence of same pattern) across 15 preparations. **(B-D)** Fictive behavioural profile of three representative preparations. Bi) Representative traces of Ca²⁺ activity in one preparation. T3 difference traces (black line) represent thoracic asymmetries. Coloured dots show incidence of different types of activity patterns. Percentage incidences of each activity pattern across the full (30 min) measurement window in stacked bar charts to the right of traces. Bii) Probability matrix of transitions between one activity pattern (preceding) to the subsequent motif (following). Biii) Kernel density plot quantifying all points of difference trace in T3 demonstrating fictive asymmetric bias within a preparation. **(C,D)** Same as in (B) for two additional preparations. **E)** Stacked bars displaying the percentage proportion of each activity pattern across all preparations. **(F)** Stacked bars displaying the frequencies of each activity pattern across all preparations. **(G)** Mean probability matrix of transitions across all preparations (N = 15). **(H)** Mean kernel density plot quantifying all points across all preparations in T3 demonstrating mean fictive asymmetric bias across all preparations. See S1 and S2 Figs for kernel density plots and probability transition matrices across all preparations. 10.17630/779141ce-c26a-483b-bfee-4f12cf71d7b2.

represent variability in cycle periods of continuous bouts of a particular motor program, but rather give an overview of how each preparation divides time amongst different particular motor programs. Different preparations demonstrated distinct biases towards certain activity patterns or combinations of behaviours. For example, some preparations showed a bias towards forwards waves (Fig 3B), others towards backwards waves (Fig 3C, 3D), or fictive head-sweeps in one direction or the other (Fig 3Ci, 3Ciii). Most preparations, however, showed at least some examples of most activity patterns (Fig 3E, 3F), though anterior bursting activity was the rarest.

Across preparations, the probabilities of transitioning among activity patterns varied widely. In most preparations, there was a low probability of repeating a given motor pattern once executed. When transition matrices from all preparations were averaged, some consistent features were apparent. In preparations biased towards backwards waves, the probability of transitioning from head-sweep to backwards wave or vice versa was high (>50%) (S2 Fig).

In our initial analyses, we used a set of criteria to identify fictive head sweeps as discrete events, as has been done in previous publications [17]. This approach identifies discrete moments when bilateral asymmetries exceed a fixed threshold. However, we noted that more subtle asymmetries were also present. To characterise these, we subtracted left from right normalised calcium signals in T3 thoracic regions, then calculated a kernel density estimation (KDE) smoothed with a Gaussian filter over the time course for each (see Methods for details) (Fig 3Biii, 3Ciii, 3Diii). The width of these plots represents the spectrum of asymmetries present and the amplitude is reflective of how prevalent asymmetric activity is overall. Each preparation showed a unique power density profile, with some preparations showing consistent bias towards one side, and others showing relatively even left-right biasing, and other preparations showing relatively little bilaterally asymmetric behaviour overall (S1 Fig). When averaged over all 15 preparations, the power density profile was relatively evenly distributed across left and right sides.

From these analyses, we can conclude that any computational model of the larval CNS should be able to generate different ratios of motor programs within and across preparations, with instantaneous frequencies of individual programs ranging from 0–0.4 Hz (0–24 per minute) and overall frequencies (i.e., events per 30 min) in a range from 0–0.08 Hz (1–5 events per minute). A model should also be able to produce variations in transition probabilities across animals and show a gradation of bilateral asymmetries in thoracic and anterior abdominal regions.

## Spontaneous deletions of wave-like activity reset forwards wave rhythm generation

In addition to wave-like activity and bilaterally asymmetric activity, isolated CNS preparations also produced anterior and posterior bursts. As noted above, these events often occurred during transitions between motor programs. However, bursting in distal regions also sometimes occurred within continuous bouts of a single type of wave-like activity (Fig 4A). These interruptions were effectively spontaneous 'deletions' of a wave within a train of similar waves. In other motor systems, similar types of spontaneous deletions have been used to assess the extent to which modulations of motor patterns can reset underlying rhythms [27]. To examine whether wave rhythm and wave pattern generation are interdependent, we measured the cycle periods of bursting in A8 for several cycles before, then directly after a spontaneous forwards wave deletion within a bout of forwards waves (see Methods for details). After a wave deletion, the A8 cycle period directly

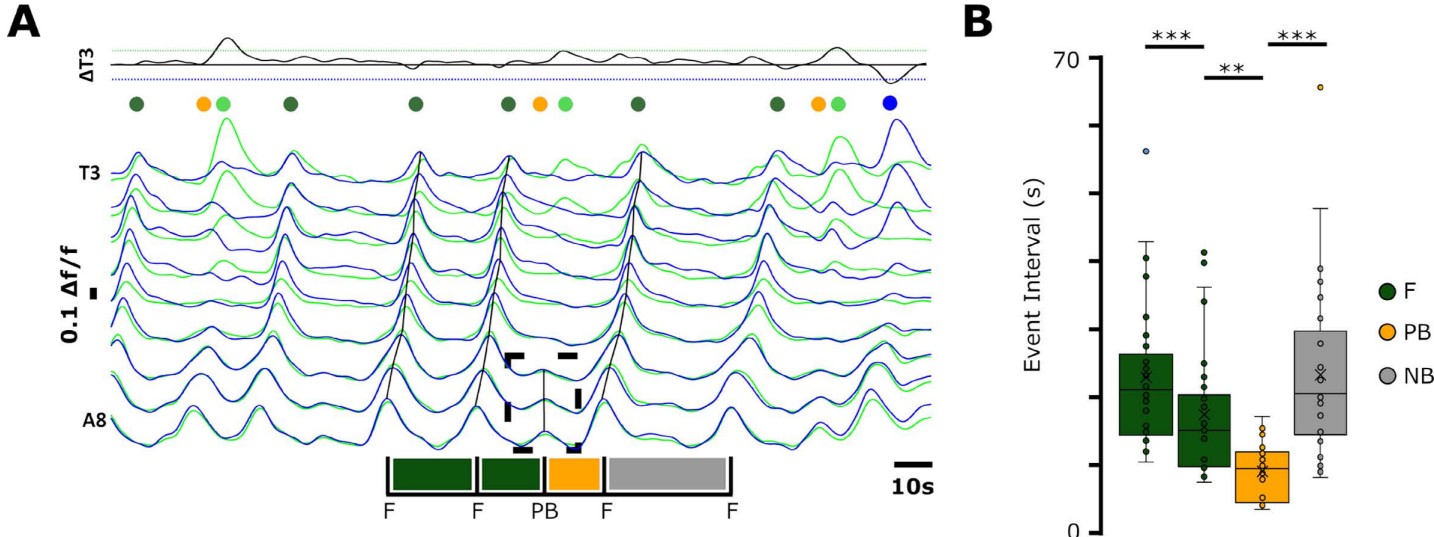

**Fig 4. Posterior bursts reset the rhythm underlying forwards wave generation. (A)** Example of posterior burst (dashed box) occurring during a bout of forwards waves (black lines). Traces annotated to show peak-to-peak cycle period of bursting in A8, showing forwards waves (marked 'F') and one posterior burst (marked 'PB'). Bar beneath traces indicates peak times in A8 and intervals between activity patterns. Note shortened cycle period after PB event. **(B)** Peak-to-peak cycle period in A8 for sequences like that shown in (A) including time to next immediate behaviour (NB). Sample size: 31 sequences across 6 preparations. * = p < 0.05, ** = p < 0.01, *** = p < 0.001. Repeated measures ANOVA, paired samples tests. 10.17630/779141ce-c26a-483b-bfee-4f12cf71d7b2.

following the deletion decreased significantly compared to preceding cycles; A8 cycle periods then lengthened once forwards wave activity was restored (Fig 4B; n = 31 sequences across 15 preparations). Intra-bout deletions of backwards waves were not frequent enough to perform the same analysis. Overall, these experiments showed that spontaneous deletion of the forwards wave pattern resets forwards wave rhythm generation, resulting in a phase advance of forwards wave initiation.

## Motor program diversity in reduced preparations

The intact isolated CNS generates multiple motor programs spontaneously. Previous work has also noted the presence of bursting activity in highly reduced preparations consisting of only a few posterior segments [26]. To examine further what types of activity patterns small numbers of segments can generate, we imaged activity in preparations comprising only segments A6-A8. We imaged activity in glutamatergic neurons, then characterised the types of motor programs each preparation produced (see Methods for details). Clear wave-like activity was interspersed with other types of irregular activity, including synchronous or near-synchronous bursting events across all hemisegments (Fig 5A). In other instances, rhythms were present but not well coordinated across segmental boundaries or across the midline. We noted the presence of subtle bilateral asymmetric activity that was synchronous across the midline, but slightly higher on one side. We also noted the presence of 'drop out' events where one or several hemisegments were recruited synchronously, but all other segments failed to recruit. In contrast, in intact CNS preparations, there was a high degree of synchrony within A6-A8 hemisegments (Figs 1–3). Overall, when removed from the rest of the CNS, individual hemisegments within a small contiguous group showed an ability to generate rhythmic activity at different frequencies simultaneously, suggesting that the underlying rhythms in these reduced preparations were not generated by a single overarching pacemaker, but rather by oscillatory circuits intrinsic to hemisegments. However, intersegmental and bilateral coordination was clearly disrupted when compared to more intact preparations containing abdominal, thoracic and brain regions.

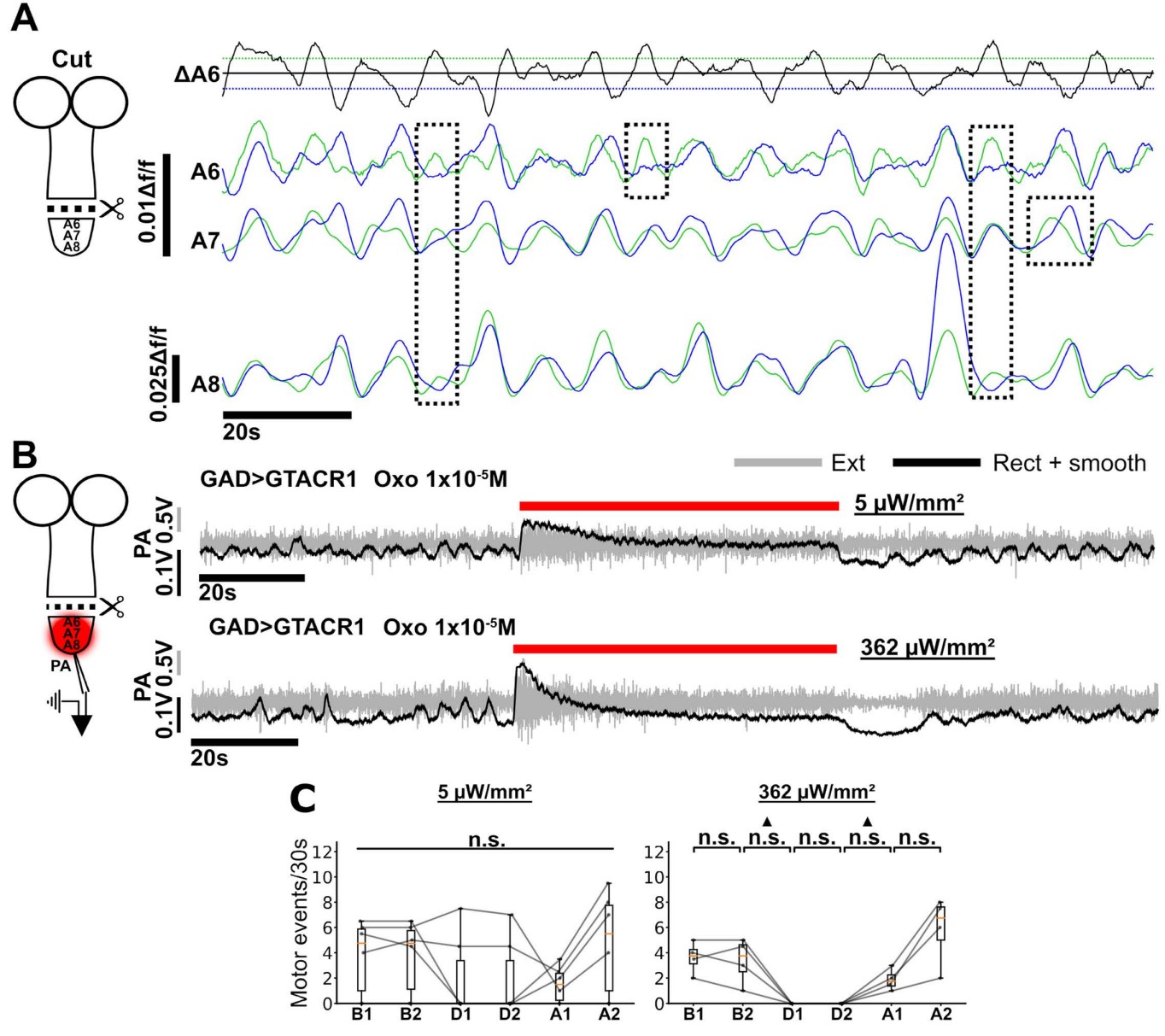

**Fig 5. Activity in GABAergic neurons is required for rhythm generation in hemisegmental oscillators. (A)** Left: Schematic of central nervous system (CNS) and the surgical removal of posterior segments (approximately segments A8 to A6). Right: Example of rhythmic activity in isolated posterior segments. ΔA6 is a computed trace of A6R (Blue) subtracted from A6L (Green). Note a different scale bar for A6 and ΔA6 traces. Dashed boxes indicate prominent examples of intra- and intersegmental asymmetric activity suggesting presence of multiple oscillators. **(B)** Electrophysiological recordings from a posterior abdomen motor nerve root from isolated posterior segments (approximately segments A8 to A6). Grey shows raw trace, black shows rectified, low pass filtered data. The light gated anion channel GTACR1 is expressed in GABAergic neurons under control of GAD-GAL4 (see Methods). Red bar indicates time of light pulse; light intensity shown at right. The muscarinic agonist oxotremorine $1\times10^{-5}$M was applied and light pulses were performed 10 min after oxotremorine application. **(C)** Burst frequency was measured in 30s time windows before, during, and after light pulses. Bursting was present before (B1,2) and after (A1,2), but not during (D1,2) light pulses, except for two preparations at 5 µW/mm². Fried-man test and Dunn post-hoc test with Sidak correction did not reveal significant differences amongst time points; however ▲ indicates significance from a Mood's median test (see text for details). n.s. = not significant, * or ▲ = $p < 0.05$, ** or ▲▲ = $p < 0.01$, *** or ▲▲▲ = $p < 0.001$ for all statistical tests. 10.17630/779141ce-c26a-483b-bfee-4f12cf71d7b2.

## Activity in GABAergic neurons is required for rhythm generation in hemisegments

Previous work has demonstrated that signalling through excitatory muscarinic acetylcholine receptors (mAChRs) is required for rhythm generation in reduced A6-A8 preparations [26]. To examine whether or not inhibition is also required for rhythmogenesis in these reduced preparations, we optogenetically hyperpolarized GABAergic neurons and recorded electrophysiological activity from motor nerve roots using suction electrodes in preparations (see Methods for details). In these experiments, we promoted rhythmic activity by bath applying the muscarinic agonist, oxotremorine [26]. Optogenetic hyperpolarization of GABAergic neurons using the light gated anion channel GtACR1 led to an increase in tonic motor neuron activity and a complete collapse of rhythmic motor activity, even at very low light levels (n = 4 preparations). A Mood's median test comparing bursting revealed significant differences in motor activity between control and experimental conditions (Fig 5B). In all preparations, these effects were reversible, such that rhythmic activity returned after removing the optogenetic disinhibition. These experiments suggest that GABAergic neurons play a critical role in enabling rhythm generation within local oscillators in abdominal hemisegments.

## Network model architecture

After examining activity patterns in experimental preparations, we set out to develop a computational model that synthesises these findings with the prior work of others. The concept base for our locomotion model derives from the abstract firing-rate model developed by Gjorgjieva and colleagues (2013) [21] which used a Wilson-Cowan (WC) formalism [28]. Our aim was to build on this by moving from a population-level dynamical system to a cellular-level network of single neurons based on a conductance-based formalism. We sought to build a parsimonious model capable of generating quantitatively similar diverse and variable output as observed in the larval system (Figs 1–4). Importantly, we made no attempt to build a circuit using full connectomic details as attempted in previous studies [12]). Our model is therefore an abstraction from the larval system, but incorporates known circuit motifs. All the neurons in our model are physiologically identical, with standard resistor-capacitor passive properties. The neurons are non-spiking [29], thereby having no HH-type $Na^+$ or $K^+$ channels, but they each have voltage-dependent $Ca^{2+}$ channels. These $Ca^{2+}$ channels have two functions. First, they allow the neurons to generate slow plateau-like potentials in response to excitatory input. This non-linear response plays a role equivalent to mutual re-excitation in a WC oscillator. Second, they provide $Ca^{2+}$ inflow to regulate synaptic transmission. All neuronal interactions are mediated by graded, non-spiking synapses [30], in which the transmitter release rate is a function of pre-synaptic intracellular calcium concentration (see Methods).

## Conditional oscillators in each hemisegment

Previous work has demonstrated that signalling through excitatory muscarinic receptors is required for rhythm generation in reduced preparations containing only posterior segments [26]. Furthermore, our optogenetics experiments suggested that rhythmic activity in posterior segments collapses when GABAergic neurons are hyperpolarised. To date, there is no evidence for neurons with intrinsic pacemaker-like properties in this system. Therefore, we kept the WC concept of oscillations produced by interaction between excitatory and inhibitory populations of neurons, but reified it at the cellular level. Thus, the basic oscillator consists of a single excitatory interneuron that excites a single inhibitory interneuron, which mediates negative feedback onto the excitor. This was extended by adding a second excitatory interneuron that makes identical synaptic connections to and from the single inhibitor. The two excitors can thus oscillate independently if they receive separate, temporally non-overlapping, depolarising command stimuli, but the single inhibitor oscillates when either excitor is active (Fig 6A, 6B). The notion is that one excitor ($E_F$) is responsible for driving forwards locomotion, the other ($E_B$) for driving backwards locomotion, and that oscillations are conditional upon a baseline level of excitation. This conditional oscillator motif incorporates information derived from experimental work. In the fly larva, A27h is a hemisegmental

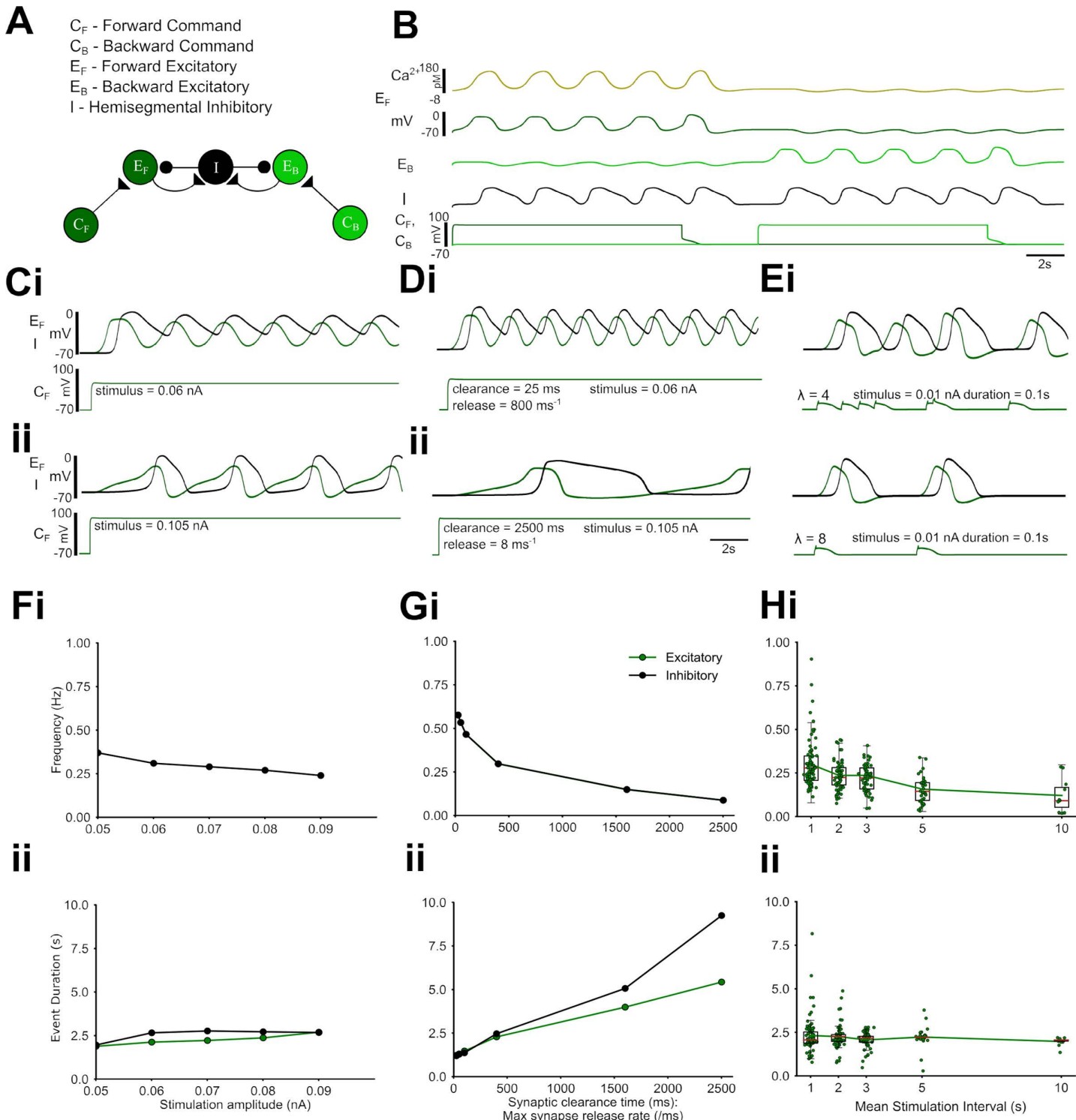

**Fig 6. The core hemisegmental oscillator (HO) can generate a range of rhythmic activity. (A)** The core HO model comprises two excitatory non-spiking neurons representing neuronal populations recruited during forwards ($E_F$, dark green) and backwards ($E_B$, light green) activity and a single inhibitory neuron (I, black). Each neuronal unit expresses a voltage-gated calcium channel. Intra-segmental excitatory and inhibitory neurons form reciprocal graded $Ca^{2+}$ dependent synaptic connections (see Methods). Command neurons ($C_F$ and $C_B$) activate the excitor for the two locomotion modes. **(B)** HO dynamics in response to tonic input to command neurons. Depolarizing a command neuron (lower traces) triggers regular oscillations in

the corresponding downstream excitor (dark green trace: $E_B$, light green trace: $E_F$) and shared inhibitor (black trace), while indirectly hyperpolarising the opposite excitor out of phase. Voltage oscillations in each neuronal unit lead to oscillations in intracellular calcium (dark yellow trace shows representative $Ca^{2+}$ oscillations in $E_B$). **(C)** $E_B$ (green) and inhibitor (black) activity in response to low (i) and higher (ii) levels of simulated current injection into $C_B$. **(D)** HO output in response to different ratios of synaptic clearance time (i, ii) to maximum synapse release rate. **(E)** HO output in response to different mean stimulation intervals (MSIs) of transient inputs drawn from a Poisson distribution and applied to $C_B$. **(F-H)** Frequency (i) and burst duration (ii) of excitor (green) and inhibitor (black) activity across a range of direct current injections into $C_B$ **(F)**, a range of ratios of synaptic clearance time to maximum release rate **(G)**, and a range of MSIs delivered to $C_B$ **(H)**. 10.17630/779141ce-c26a-483b-bfee-4f12cf71d7b2.

excitatory interneuron that is only active in fictive forwards locomotion [31], while A18b is a hemisegmental excitatory interneuron that is only active in fictive backwards locomotion [32], and GDL is a hemisegmental inhibitory interneuron that is active in both locomotor modes [31,32]; GDL is just one of multiple local inhibitory interneurons, but stands as exemplar of a shared inhibitor in this system. While it is not implied that the larval system ensures complete segregation of exciters between forwards and backwards modes, nor that they fully share inhibitors, these types motifs are known to exist in vivo. It was plausible, therefore, to incorporate them into our simplified network model. Thus, the approximately 120 neurons per hemisegment involved in generating oscillations underlying forwards and backwards locomotion in fly larvae are condensed into just 3 neurons in our model.

Together, the three neurons constitute the hemisegmental oscillator (HO). The dynamics of oscillation depend on the strength of the command input and the kinetics of the neuronal and synaptic properties constituting the HO (Fig 6A). The command input is both necessary and sufficient to trigger oscillations and therefore meets accepted criteria for command neurons proposed by [33]. Values were chosen so that the system is silent at rest, but maintains oscillations in response to tonic exciter stimulation. Varying the strength of the command input changes the output frequency over an approximately 2-fold range, but this can be increased to an approximately 10-fold range by also altering synaptic transmitter release and clearance rates (Fig 6C, 6D, 6F, 6G). This approximates the tonic frequency variability seen in fictive locomotion in fly larvae (e.g., Fig 3; [26]. Single brief command pulses applied to the HO produce single cycles of oscillation that outlast the pulse, while stochastic pulses with a Poisson distribution of interpulse intervals can generate oscillations with cycle-by-cycle frequency variability that again spans the range seen in the larval system, but in this case without requiring any change in synaptic properties or command strength (Fig 6E, 6H).

### Architecture of the abdominal model

The HO was replicated eight times to model the chain of abdominal ganglia on one side of the animal (Fig 7A). In the fly larva, feed-forwards excitation between segments mediated by pre-motor excitatory interneurons acts as a delay circuit that helps generate the necessary phase lag for metachronal progression [34]. In the model, metachronal progression of the locomotion wave is ensured by direct feed-forwards excitation from the excitatory HO interneuron in the active segment to both the excitatory and inhibitory HO interneurons in the next segment recruited in the wave (anteriad for forwards waves, posteriad for backwards waves: $E_{F\_n} \rightarrow E_{F\_n+1}$, $E_{F\_n} \rightarrow I_{n+i}$, $E_{B\_n} \rightarrow E_{B\_n-1}$, $E_{B\_n} \rightarrow I_{n-i}$). The relatively slow dynamics chosen for the non-spiking synapses obviates the need for a separate delay mechanism.

### Metachronal wave initiation and progression

In the model, a metachronal wave is initiated by stimulating command neurons (C) that mediate local excitation directed either to HO excitors in the three posterior ganglia to generate forwards waves ($C_F \rightarrow E_{F\_posterior}$), or to those in the two anterior ganglia to generate backwards waves ($C_B \rightarrow E_{B\_anterior}$) (Fig 7Bi, 7Bii). During fictive locomotion in fly larvae, the group of 2–3 terminal ganglia often show activity peaks at the same time (Fig 2, [12]). In the model, a similar effect was produced by placing multiple terminal ganglia downstream of single command neurons (e.g., compare waves in Figs 2A and 7B). Concurrent generation of forwards and backwards locomotory waves is prevented because the command neurons also

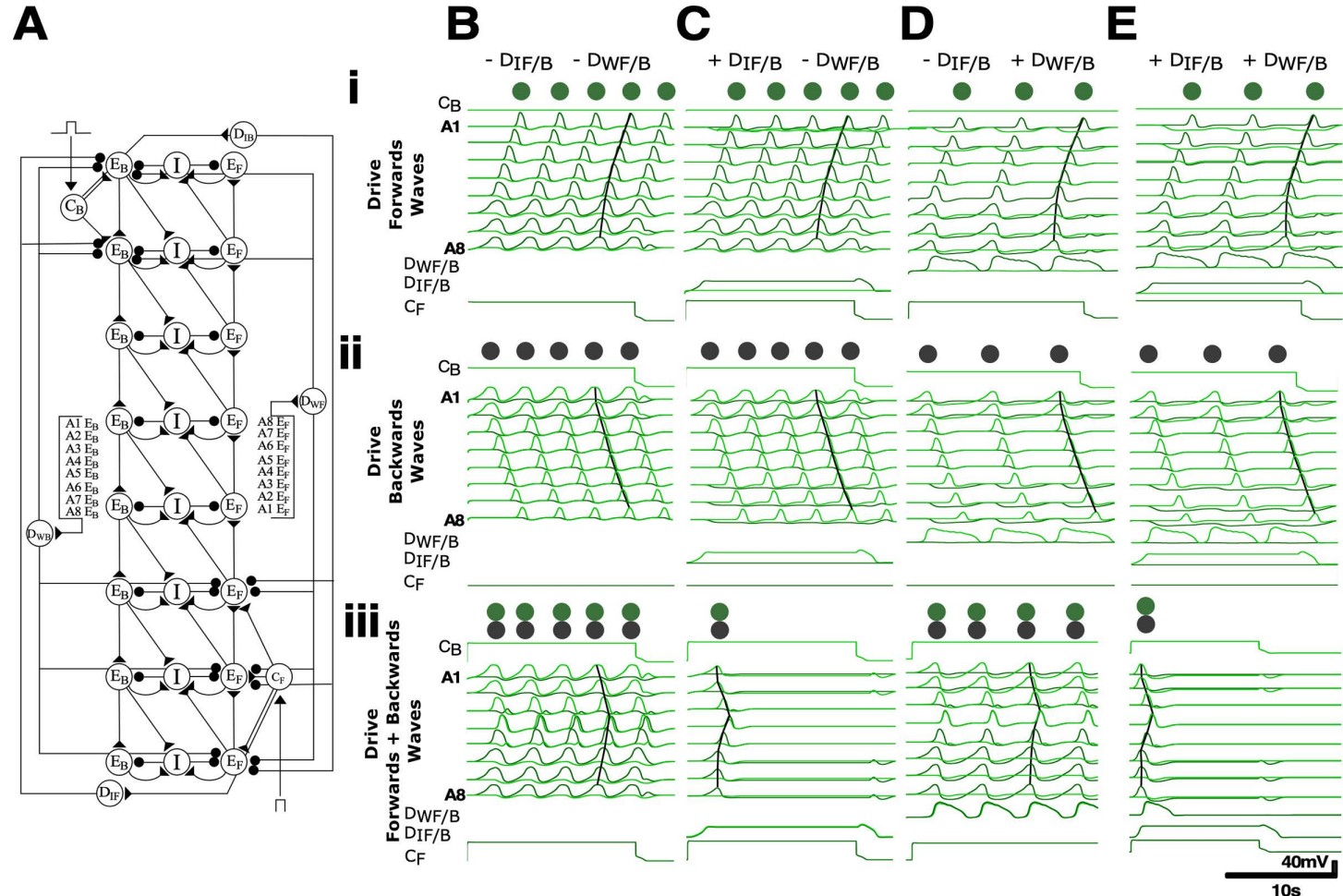

**Fig 7. Wave initiation detector motifs primarily segregate opposing motor programs while wave detector motifs segregate both opposing and same motor programs. (A)** Initiation detectors ($D_{IF/B}$) are innervated by opposite command neurons and locally inhibit initiating excitors (A6-A8 or A1-A2). Wave detectors $D_{WF/B}$ receive excitatory inputs from each segment, and inhibit generation of the same and opposite motor programs. In the schematic, triangles indicate excitatory connection, circles indicate inhibitory synaptic connection. Note all inhibitory synapses from wave and initiation detectors have maximal conductance that are twice as large as all other inhibitory synapses. **Bi-iii)** Voltage traces show the effects of tonically exciting (i) posterior command neurons ($C_F$) driving forwards waves, (ii) anterior command neurons ($C_B$) driving backwards waves, and (iii) both command neurons simultaneously in circuits lacking both $D_{IF/B}$ and $D_{WF/B}$. Note overlap in timing of forwards and backwards waves. **Ci-iii - Ei-iii)** Same as in (B) but with $D_{IF/B}$ neurons and not $D_{WF/B}$ neurons present **(C)**, $D_{WF/B}$ neurons and not $D_{IF/B}$ neurons present **(D)** and with both $D_{IF/B}$ and $D_{WF/B}$ neurons present **(E)**. Overall, $D_{IF/B}$ are necessary for segregating opposite motor programs (compare Biii and Ciii) while $D_{WF/B}$ are required for segregation of same motor programs (compare Ci-ii and Di-ii). 10.17630/779141ce-c26a-483b-bfee-4f12cf71d7b2.

activate initiation detector neurons ($D_{IF}$ and $D_{IB}$, for forwards and backwards, respectively) which inhibit the opposite-mode excitatory interneurons at the other end of the oscillator chain ($C_F \rightarrow D_{IF} \nrightarrow E_{B\_anterior}$, $C_B \rightarrow D_{IB} \nrightarrow E_{F\_posterior}$, where crossed-out arrows indicate inhibition, Fig 7Cii).

In fly larvae, paired 'mooncrawler' descending interneurons (MDNs) located in the brain have command-like functions and, when stimulated, initiate backwards locomotion and terminate any concurrent forwards locomotion. They activate A18b interneurons ($E_B$ equivalent) in anterior segments; they also activate GABAergic Pair 1 descending interneurons which inhibit A27h interneurons ($E_F$ equivalent) in posterior segments [32]. Our $C_B$ and $D_{IB}$ neurons thus replicate in part the functionality of the larval MDN and Pair 1 interneurons. Relatively little is known about the neural circuitry initiating

forwards locomotion in the larval fly, so in the model, we chose to simply duplicate the MDN and Pair 1 motifs, instantiated as $C_F$ and $D_{IF}$ neurons. Conceptually, $D_{IB}$ and $D_{IF}$ neurons serve to detect command inputs that initiate and/or sustain motor programs and then project to regions where opposing motor programs arise.

In this configuration, a brief command stimulus initiates a single locomotory wave, but sustained command activation causes repeated waves of activity, as is often seen in the larval system. However, there is no constraint preventing initiation of a second wave before completion of the first wave (Fig 7Bi, 7Bii). In fictive locomotion in the larval system, this overlap does not normally occur. This implies that something prevents the initiation of the second wave while the first wave is still in progress. In our model, this constraint is implemented by extrasegmental wave detectors ($D_W$) that are excited by HO excitors in each segment ($E_F$s and $E_B$s), and thus are active throughout wave progression. These neurons inhibit excitors in the terminal ganglia at either end of the chain, thereby reducing the likelihood of a new wave initiating while the preceding wave is still in progress (Fig 7Cii). There are separate detectors for forwards and backwards waves ($D_{WF}$ and $D_{WB}$, respectively). In fly larvae, it is clear from calcium-imaging data that there are several neuron types that are active throughout wave progression [17] and some that are recruited specifically during waves in one direction [35], and these could act as wave detectors. Furthermore, recent work suggests that interneuron populations that monitor motor activity could be involved in preventing maladaptive overlap of motor programs [36].

### Interrupted wave progression

In fictive locomotion in the larva, it is fairly common for a wave to abort—i.e., to initiate, but then fail to progress along the entire length of the nerve cord. Abortive forwards waves are termed here and in previous publications as 'posterior bursts' and abortive backwards waves are termed 'anterior bursts'. Posterior bursts in isolated preparations are characterised by a phase advance in subsequent complete waves, suggesting a network reset that affects the phase of subsequent events across the entire abdominal chain, not just the segments downstream from the point of interruption (Fig 4). In larvae, both forwards and backwards wave progression can be halted by specific optogenetic activation of the inhibitory hemisegmental GDL interneuron within a small section (2–3) of adjacent mid-abdominal segments [31], although there is no evidence that adventitious GDL activation is the causal mechanism underlying the spontaneous abortive waves.

In the computational model, wave progression can be halted mid-chain by inhibiting the appropriate segmental E neuron ($E_F$ to block a forwards wave, $E_B$ to block a backwards wave), or by exciting the appropriate I neuron (which blocks waves in either direction). Furthermore, when this block is applied, there is a phase advance of subsequent waves that affects the entire chain (Fig 8). The latter effect is a direct consequence of the wave detector neurons ($D_{WF,B}$). These sum activity over the whole chain, so if wave progress is blocked part way, the wave detector neurons receive excitation over a shorter time period, and consequently mediate briefer inhibition onto the initiating segments, leading to a phase advance for the next cycle. The further the wave progresses before it is blocked, the less change in the duration of $D_W$ activation there is, and the less of a phase advance there is on initiation of subsequent waves. This architecture leads to a positive correlation between wave duration and burst duration for forwards ($R^2 = 0.73$) and backwards waves ($R^2 = 0.95$), respectively (Fig 8C). If wave detectors are completely removed from the circuit, then wave progression is still blocked by the mid-chain E/I manipulation, but there is no effect on the oscillator phase of subsequent waves when the block is removed.

### Stochastic inputs to command neurons promote motor program switching

In freely moving, intact larva, the animal usually produces sustained bouts of forwards locomotion, which are interrupted by backwards locomotion or bilaterally asymmetric head-sweeps if the animal encounters an obstacle. In fictive locomotion produced by the isolated larval CNS, all three motor programs are represented, but forwards waves do not dominate in all preparations (Fig 3, [24]. There are sometimes sustained bouts of one particular mode but in many preparations there are frequent switches in the direction of locomotion and in the occurrence of abortive waves (anterior and posterior

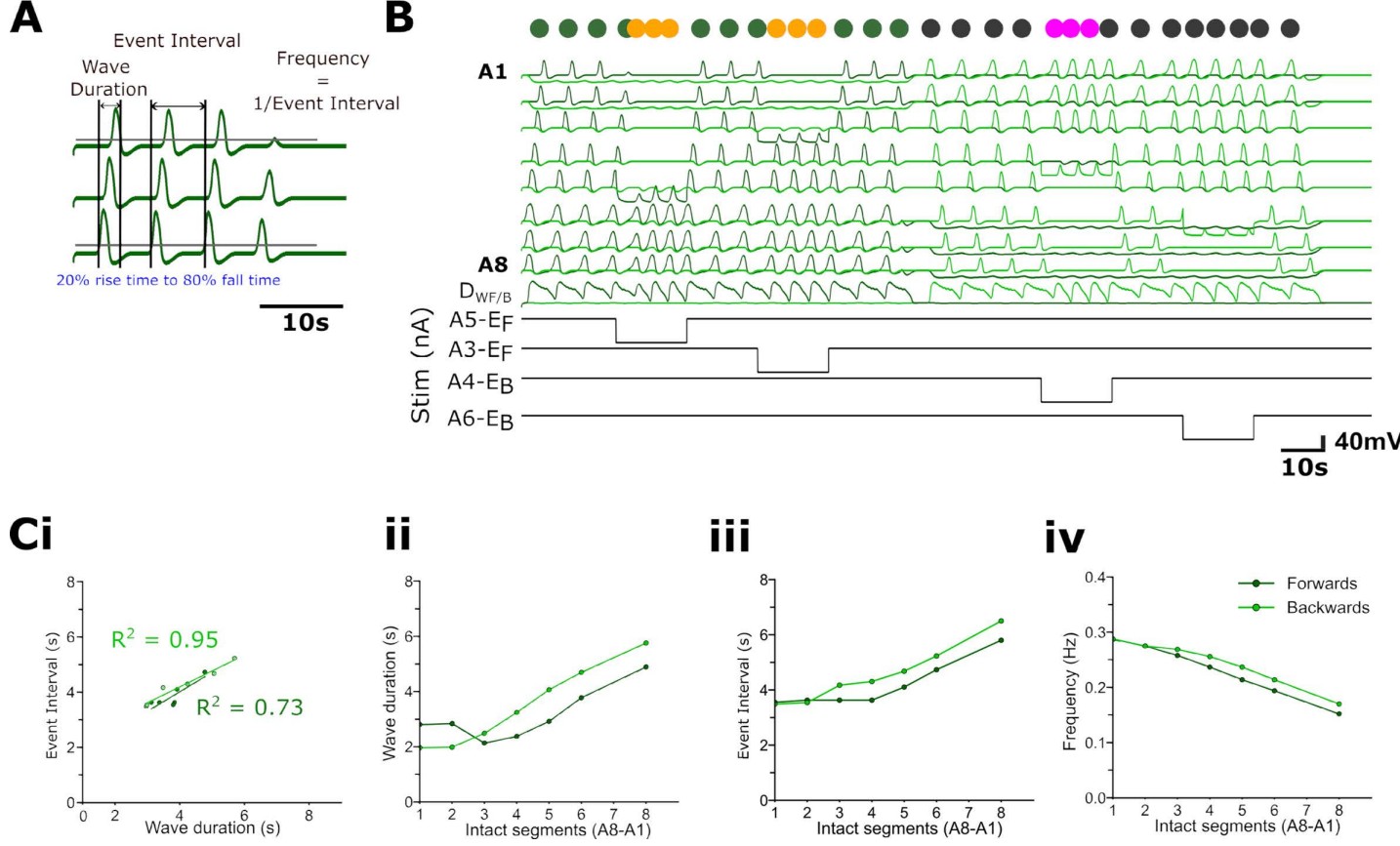

**Fig 8. Interrupting wave progression resets rhythm generation in abdominal model incorporating wave detectors and initiation detectors.** Inhibiting exciters ($E_F$ or $E_B$) blocks forwards and backwards wave propagation across segments in a stepwise fashion, thereby effectively reducing the number of intact segments. **(A)** Quantification of wave duration as the time between 20% of the rising phase of activity in the first segment in the chain to 80% of the falling phase of activity in the last segment. Event interval is the time between equivalent points in consecutive cycles. Frequency is the inverse of the event interval. **(B)** Examples of decreases in event interval during forwards and backwards waves at two levels of impaired wave propagation; wave duration decrease is apparent and is proportional to the number of segments before the activity block. **(C)** Linear relationships between wave duration and cycle period; $R^2$ values shown for forwards (light green) and backwards (dark green) waves. **(Cii-iv)** Wave duration (s) (ii), burst interval (s) (iii) and wave frequency (Hz) (iv) depend on the number of intact (unblocked) abdominal segments before the activity-blocked segment. 10.17630/779141ce-c26a-483b-bfee-4f12cf71d7b2.

bursts) and asymmetries (Fig 3). This suggests that the command systems activating the various modes are less stable in the isolated system than in the intact system, which presumably is due to the loss of stabilising proprioceptive and exteroceptive feedback.

To simulate the input of unpatterned information into the model network, command neurons were stimulated with brief (0.1 s, 0.01 nA) stochastic current pulses in which the intervals between subsequent stimuli were derived from a Poisson distribution. This distribution is advantageous for capturing randomly occurring events, each with a stochastic number of inputs to the command system. Such a distribution would arise if a command system with a threshold were activated by random noise, rather than by coordinated inputs. Under these conditions, the network generates a mixture of four main output patterns: forwards and backwards waves (Fig 9Aii, 9Aiii), and abortive anterior and posterior bursts without collisions (Fig 9Ai,iii,iv). These patterns were qualitatively similar to those seen in the larva (Figs 2-3). Activity in Fig 9Ai is reminiscent of 'see-sawing' that can occur in larval fictive locomotion (Fig 2), in which an alternating succession of anterior and posterior bursts fail to propagate into metachronal waves, while Fig 9Aii shows forwards wave propagation. Phase

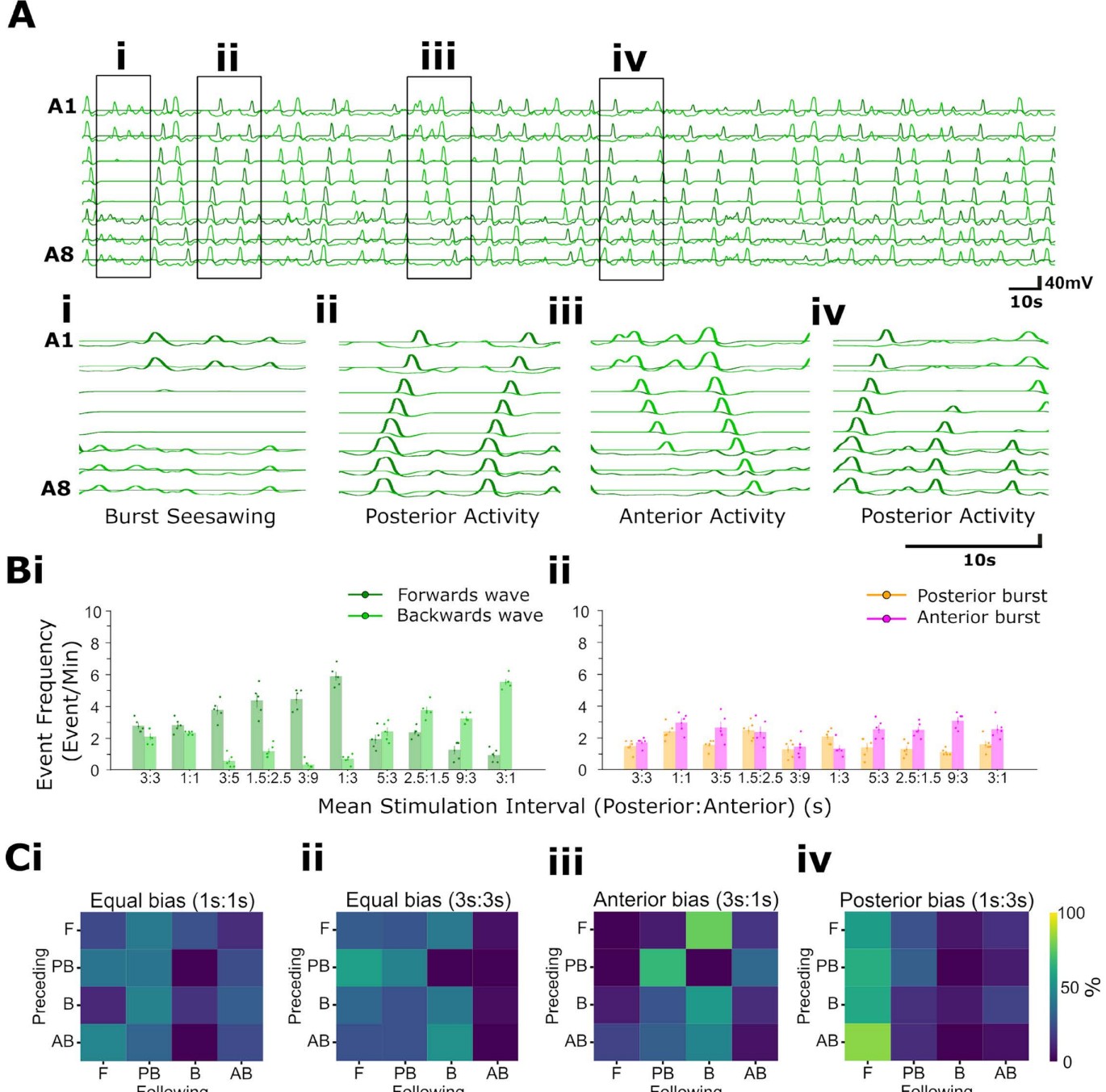

**Fig 9. Stochastic command inputs to the abdominal model reproduce irregularities observed in experimental preparations. (A)** The output of the abdominal model with equal anterior (stimulating $C_B$) and posterior (stimulating $C_F$) mean stimulus intervals (MSI). Light green traces: $E_B$ voltage; dark green traces: $E_F$ voltage. The model exhibits bouts of abortive activity in distal regions (i); forwards waves (ii); backwards waves and anterior bursts (iii); and posterior bursts (iv). **(Bi)** Event frequency per minute of forwards and backwards waves with varying ratios of MSI delivered to posterior ($C_F$) and anterior ($C_B$) command-like neurons (i). **(Bii)** Event frequency per minute of anterior and posterior bursts through the same range of posterior:anterior MSIs. **(Ci-iv)** Transition matrix heatmaps for (i) 1s:1s equal bias, (ii) 3s:3s, (iii) 3s:1s anterior bias and (iv) 1:3s posterior bias. Colour represents the proportion of total transitions composed by the transition. Data are pooled from 2 simulation runs at each MSI ratio. 10.17630/779141ce-c26a-483b-bfee-4f12cf71d7b2.

advances following an anterior burst were also apparent (Fig 9Aiii), confirming that rhythm resetting following an aborted wave can arise with non-tonic stimuli similar to the resetting seen when metachronal progression is deliberately interrupted in waves driven by tonic stimuli (Fig 8).

In the Poisson distribution, the timing of stimulus occurrence is defined by a single parameter, the mean stimulus interval (MSI) between events (thus a larger interval leads to a lower average stimulus frequency). When the MSI applied to $C_F$ and $C_B$ was the same, the four output patterns occurred with approximately equal frequency, albeit with a slight bias towards forwards waves compared to backwards waves (Fig 9B). This bias is likely due to the asymmetry in the number of anterior segment $E_B$s (2) compared to posterior segment $E_F$s (3) receiving command activation (Fig 7A) targeted by their respective command neurons. There was little change in wave output frequency when the MSI ratio was decreased from 3:3 s to 1:1 s. At the higher frequency, many activations of command neurons occur while a wave is already in progress, and initiation of a new wave is gated by the inhibition mediated by wave detectors ($D_W$). If the MSI delivered to $C_B$ were increased without any change in MSI delivered to $C_F$, the frequency of backwards waves would be expected to decrease, given the drop in the frequency of $C_B$ activation. However, we also noted a relative increase in the frequency of forwards waves, even though the MSI $C_F$ activation was unchanged. This is a consequence of indirect mutual inhibition mediated by initiation detectors ($D_{IB/F}$). A drop in $C_B$ activation rate means that some of the $C_F$ activations that would previously have been blocked by $D_{IB}$ activation are now effective in generating forwards waves, leading to a rise in frequency of forwards waves even without an increase in $C_F$ activation rate. A similar but reversed effect was seen if the mean stimulation interval for $C_F$ was increased, with that of $C_B$ unchanged (Fig 9Bi). There are clear differences in the effects of command biasing on anterior and posterior burst generation arising from the different number of anterior versus posterior of activating segments (Fig 9Bii).

## The thoracic head sweep circuit

In addition to forwards and backwards waves, larvae often sweep their heads to scan their sensory environment to move towards desirable or away from noxious stimuli. A head-sweep to one side is often quickly followed by a sweep to the opposite side. In addition, given a sensory gradient, the direction, magnitude and frequency of head-sweeps have been shown to be altered [37,38]. The isolated CNS generates fictive head-sweeps comprising bilaterally asymmetric activity in thoracic and anterior abdominal regions (Figs 1, 3; [17]). These asymmetries are not strictly alternating, but rather irregular, with left and right sides spontaneously generating events that may alternate, occur in isolation, or show repetitive ipsilateral activity. Previous experimental work has also found that thoracic hemisegments can generate oscillations in isolation from corresponding contralateral hemisegments, suggesting that head-sweep rhythmogenesis does not strictly require connections across the midline [26].

To model fictive head-sweeps, we built a thoracic circuit by extending the concepts of the abdominal circuit. It comprised three bilateral pairs of HOs (Fig 10A), representing segments T1-T3 with random inputs given as before to command neurons on the left and right sides of the circuit, $C_L$ and $C_R$, respectively. Given that there are functional links and timing overlaps between head-sweeps and backwards waves, we postulated that $E_B$ neurons in thoracic regions could be integral to head-sweep circuitry, while also serving to trigger backwards waves. Left and right $E_B$s therefore get their input from corresponding ipsilateral head-sweep command neurons. Head-sweep detectors ($D_H$) were modelled as merged wave and initiation detectors with left and right counterparts ($D_{HL}$ and $D_{HR}$, respectively). Both are excited themselves by ipsilateral $E_B$s. $D_{HL}$ and $D_{HR}$ then project inhibitory connections onto contralateral $E_B$s in a half-centre inhibition motif. This configuration does not lead to an intrinsic head-sweep oscillator with a fixed frequency but instead allows alternation of bias towards left or right asymmetries and the generation of irregular frequencies by changing the ratios of left and right command inputs and synaptic properties in the thorax.

When head-sweep detectors ($D_{HL/R}$) were removed, and a 1:1 s MSI ratio was implemented across left and right sides, each side was able to generate rhythmic activity in isolation. This is consistent with previous published work

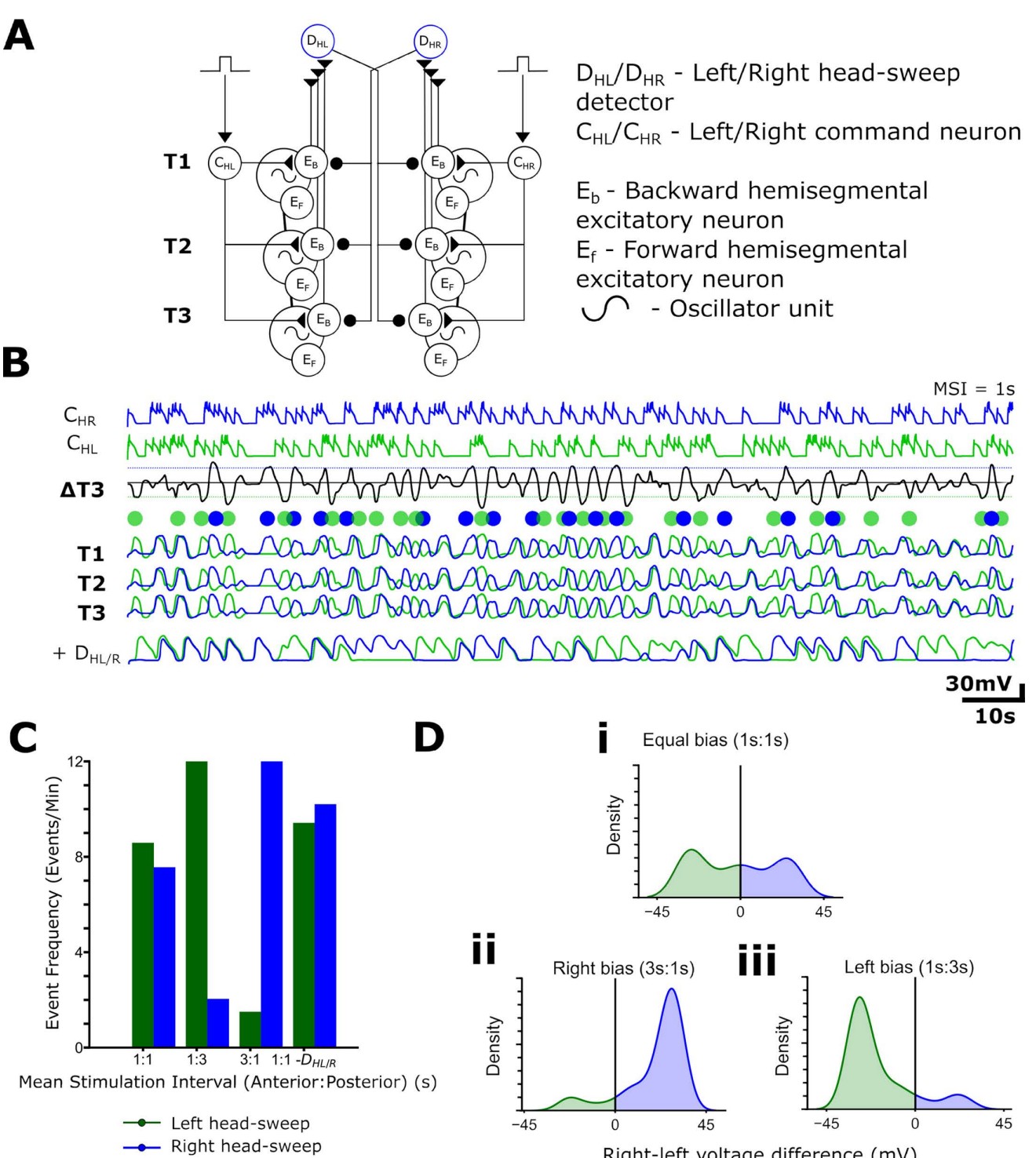

**Fig 10. Thoracic circuit model with varied left/right input probabilities recapitulates a spectrum of bilaterally asymmetric activity and head-sweeps. (A)** Circuit diagram of proposed head sweep circuitry (left) and abbreviations used (right). **(B)** Model output for 1:1 MSI ratio between $C_L$ and $C_R$. T1, T2, T3 show left (green) and right (blue) $E_B$ activity traces from the respective segment. ΔT1 is the difference trace of activity in left/right T1 $E_B$s. Dots above T1 trace indicate left (green) and right (blue) head-sweeps. **(C)** Frequency of discrete fictive head sweep events for different MSI ratios of stimuli to right and left command-like neurons. See Methods for details of criteria used for defining fictive head-sweeps. **(D)** Density plots of T1 right-left voltage difference for different MSI ratio. 10.17630/779141ce-c26a-483b-bfee-4f12cf71d7b2.

showing that thoracic hemisegments are able to generate rhythmic activity in the absence of contralateral connections [26]. With $D_{HL/R}$ present, and a 1:1 s L-to-R MSI ratio, the network produced a range of bilaterally asymmetric activity patterns (Fig 10B). The frequency of discrete left and right head-sweep events (see Methods for criteria used) was similar in 1:1 s MSI bias conditions. MSI bias to left resulted in increased frequency of left head-sweep and vice versa (Fig 10C). KDE plots of asymmetric activity revealed that there was a distribution of bilateral asymmetries, as in experimental preparations. As predicted, biasing MSI ratios towards one side resulted in a corresponding shift in the KDE plot to that side (Fig 10D).

## The full integrated network model

Next, we combined our abdominal and head-sweep circuits with the aim of creating a full model capable of reflecting fictive activity patterns seen in the isolated CNS. The abdominal ladder and associated extrasegmental circuitry was duplicated bilaterally to represent both sides of the CNS and excitors were connected within each segment to their counterparts across the midline (Fig 11). Excitatory connections from thoracic $E_F$s were fed onto $D_{WF}$ and thoracic $E_B$s onto $D_{WB}$ to ensure full segregation of motor programs. Experimentally, A1-A2 often appears to act as an integrating zone of thoracic and abdominal activity, exhibiting asymmetries from head-sweeps that often resolve into symmetric backwards waves in more posterior segments or taper off as anterior bursts (Fig 1). A synaptic chiasm between $E_B$s in L/R-A1 and contralateral L/R-A2 $E_B$s was built into the model to facilitate the initiation of symmetric waves from asymmetric thoracic activation, as observed in isolated larval CNS preparations.

Evidence for this type of contralateral, inter-segmental excitatory connection has been demonstrated in previous connectomics studies [12]. Functionally, we replaced the $C_B$ command-like neuron that triggered backwards waves with $C_{HL/R}$ (Fig 7A). This enabled backwards abdominal waves to arise from left/right head-sweeps or symmetric thoracic activity, recapitulating our experimental result.

Equal bias stimulation of $C_{FL/R}$, $C_{HL}$ and $C_{HR}$ produced a mixture of fictive activity patterns (Fig 12Ai) that resembles the equal ratios of motor programs exhibited in biological data in some preparations (Fig 12Aii). Biasing MSI ratios towards posterior regions (reducing MSI in posterior regions relative to anterior) increased the proportion of forwards waves (Fig 12Bi) which resembles the dominance in forwards waves displayed in some biological preparations (Fig 12Bii). Conversely, biasing towards anterior regions caused a marked increase in the diversity of anterior activity, including left/right asymmetries, symmetrical anterior bursts and backwards waves (Fig 12Ci), again showing similarity to preparations where right head-sweeps persist between forwards and backwards wave activity (Fig 12Cii).

By alternating the MSI ratio between the left/right anterior and posterior biases, we were able to replicate a diversity of motor event frequencies that captures the range demonstrated in fictive biological preparations (Fig 12D). Across the biased model preparations, variation extended from 2 to 4 events per minute. Each run of the model generated different ratios of fictive behaviours (Fig 12D). As expected, transition matrices for different trials showed a high probability of asymmetric events preceding backwards waves regardless of the relative MSI ratio. Biasing the network towards anterior activity by lowering the relative $MSI_{posterior}$ resulted in more transitions to backwards wave and anterior burst activity. In contrast, by biasing the network to posterior activity by raising the relative $MSI_{posterior}$, we evoked more forward waves and greater posterior burst activity (Fig 12Eii, 12Eiii).

In the course of quantifying how our model transitioned amongst different motor programs, we noted the presence of qualitative features in our model that were also present in experimental preparations (Fig 13). For example, both experimental preparations and simulations sometimes generated backwards waves followed immediately by a forwards wave, with a head-sweep occurring during the transition (Fig 13i). In addition, experimental preparations and simulations also sometimes reverberated between anterior head sweeps and posterior bursts without triggering any type of network-wide activity through all segments (Fig 13ii). Finally, sequences of multiple posterior bursts interspersed with head sweeps sometimes preceded forward waves (Fig 13iii)

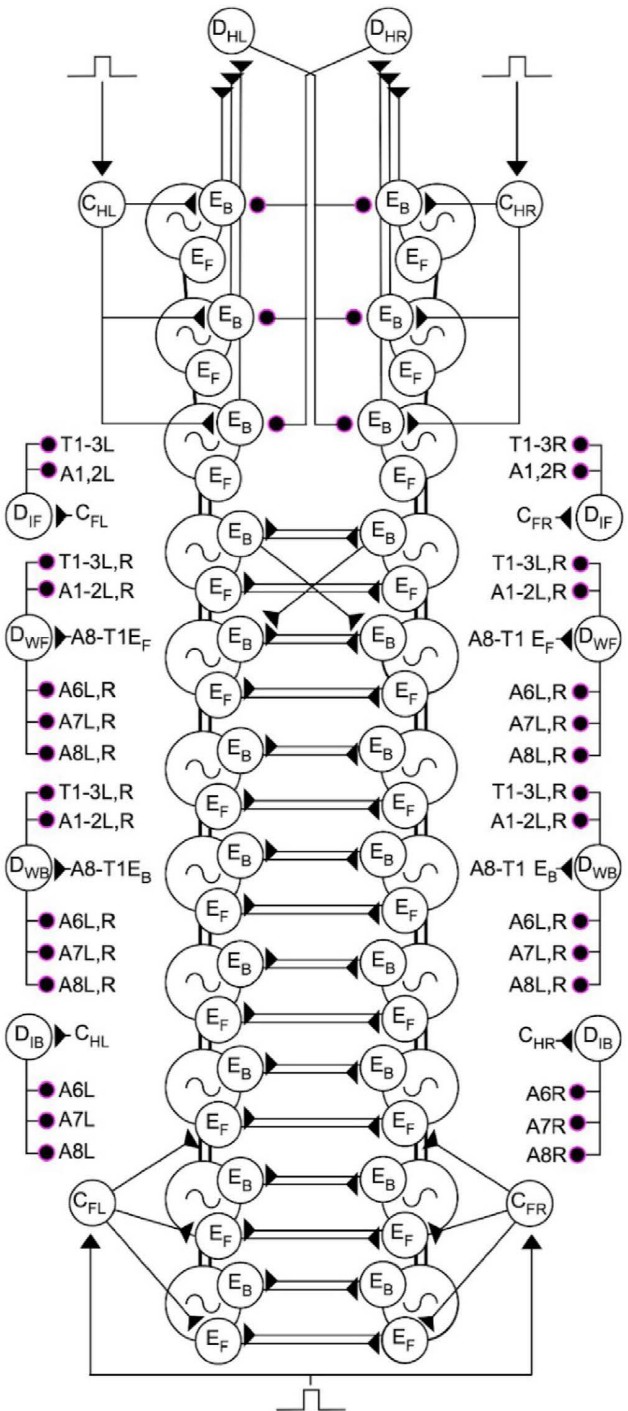

**Fig 11. Schematic of larval locomotor system model.** Abdominal circuitry including initiation and wave detectors is duplicated bilaterally. HO oscillators are represented by large circles with an oscillator symbol. Intersegmental connections are represented by black lines connecting HOs. Reciprocal intersegmental connections amongst neighbouring HOs are represented by black lines. Initiation and wave detector motifs on each side are represented outside the core model. All abdominal motifs are the same as in Fig 7. The abdominal circuit is now connected to the head-sweep circuit by connecting HOs in the thoracic T3 to A1 HOs. Note the intersegmental contralateral connections from A1-A2 and the absence of reciprocal intersegmental connections between HOs in thoracic regions (only posterior-anterior connections are present). Square wave pulse symbols represent inputs into command-like neurons. 10.17630/779141ce-c26a-483b-bfee-4f12cf71d7b2.

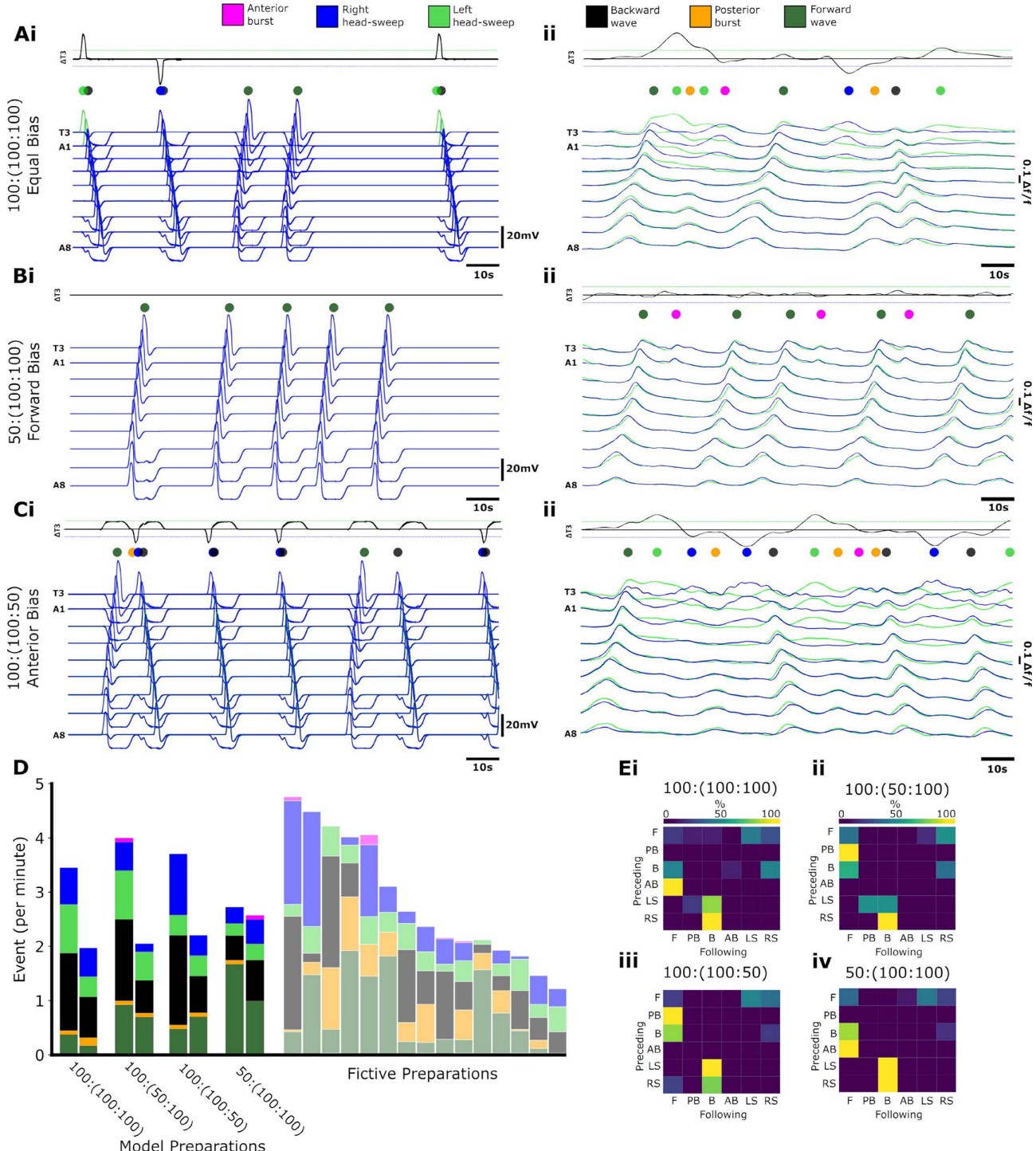

**Fig 12. The full network model can reproduce diversity and variability observed in experimental preparations.** Model **(i)** compared to calcium imaging traces **(ii)** for **(A)** equal bias (100:(100:100)) ((MSI$_{posterior}$:(MSI$_{anterior left}$: MSI$_{anterior right}$)., **(B)** forwards bias (50:(100:100)), and **(C)** anterior bias (100:(50:100)). **(D)** Event frequencies per minute for model and fictive preparations. The model stacked bar chart shows representative high- and low-level activity preparation from N = 10 model runs. The model stacked bar charts are shown relative to all biological fictive preparations (right, faded). Matrices showing transition probabilities between fictive behaviours in **(Ei)** equal bias, **(Eii)** left anterior bias, **(Eiii)** right anterior bias, and **(Eiv)** forwards bias. 10.17630/779141ce-c26a-483b-bfee-4f12cf71d7b2.

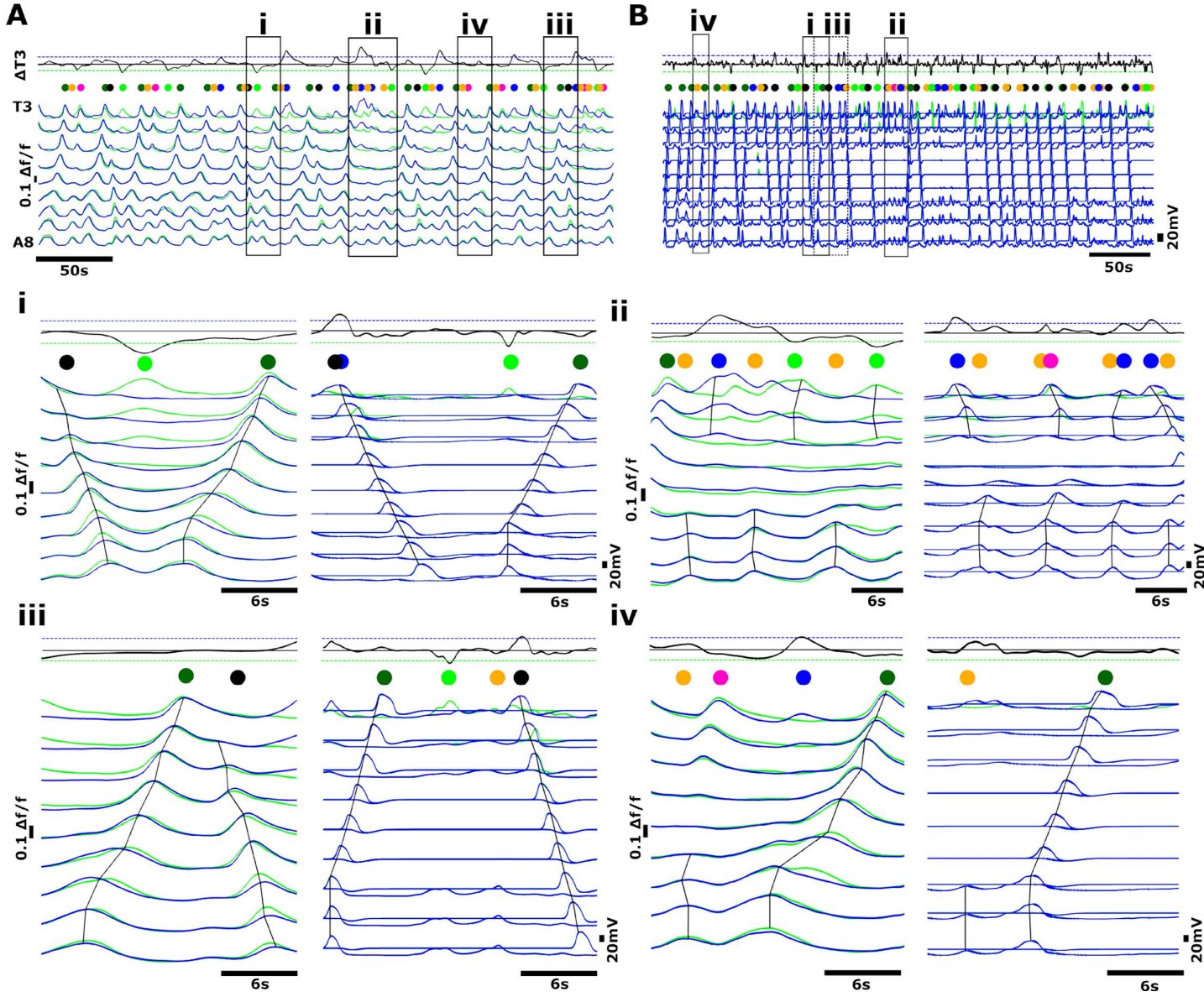

**Fig 13. Model and experimental data show qualitative similarities in complex activity sequences involving multiple motor programs. (A)** Example of rhythmic calcium signals in glutamatergic neurons in an isolated CNS preparation. Activity on the left (green) and right (blue) sides of CNS are shown. Coloured dots represent distinct motor programs, colour code same as in Fig 12. Black trace shows subtraction trace of activity across T3. Dashed horizontal lines indicate threshold for detecting head sweeps. **(B)** Model output with stochastic inputs to posterior and anterior regions. Dashed boxes indicate regions of interest. Each numeral indicates an instance of qualitatively similar activity sequence in experimental preparation and model. Produced using a 1:(1:1) ratio ((MSI_posterior:(MSI_anterior left : MSI_anterior right) (i) Backward wave followed by forwards wave with head-sweep during moment of transition. (ii) Posterior burst alternating with head sweep activity, no wave like activity. (iii) Low amplitude posterior bursts alternating with head sweeps, followed by forwards waves. Note difference in time course of calcium signals in preparations vs. time course of voltage in model. 10.17630/779141ce-c26a-483b-bfee-4f12cf71d7b2.

We also noted a major qualitative difference between experimental preparations and the model: durations of activity bursts in different hemisegments in the model were shorter than those in preparations (Fig 13, all panels). This difference in time course makes sense given that in experimental preparations, we were not recording voltage, but rather slow calcium transients in a heterogenous population of glutamatergic neurons, further transformed by the time constant of GCaMP. This is in contrast to our model where we directly measured voltage in single compartments.

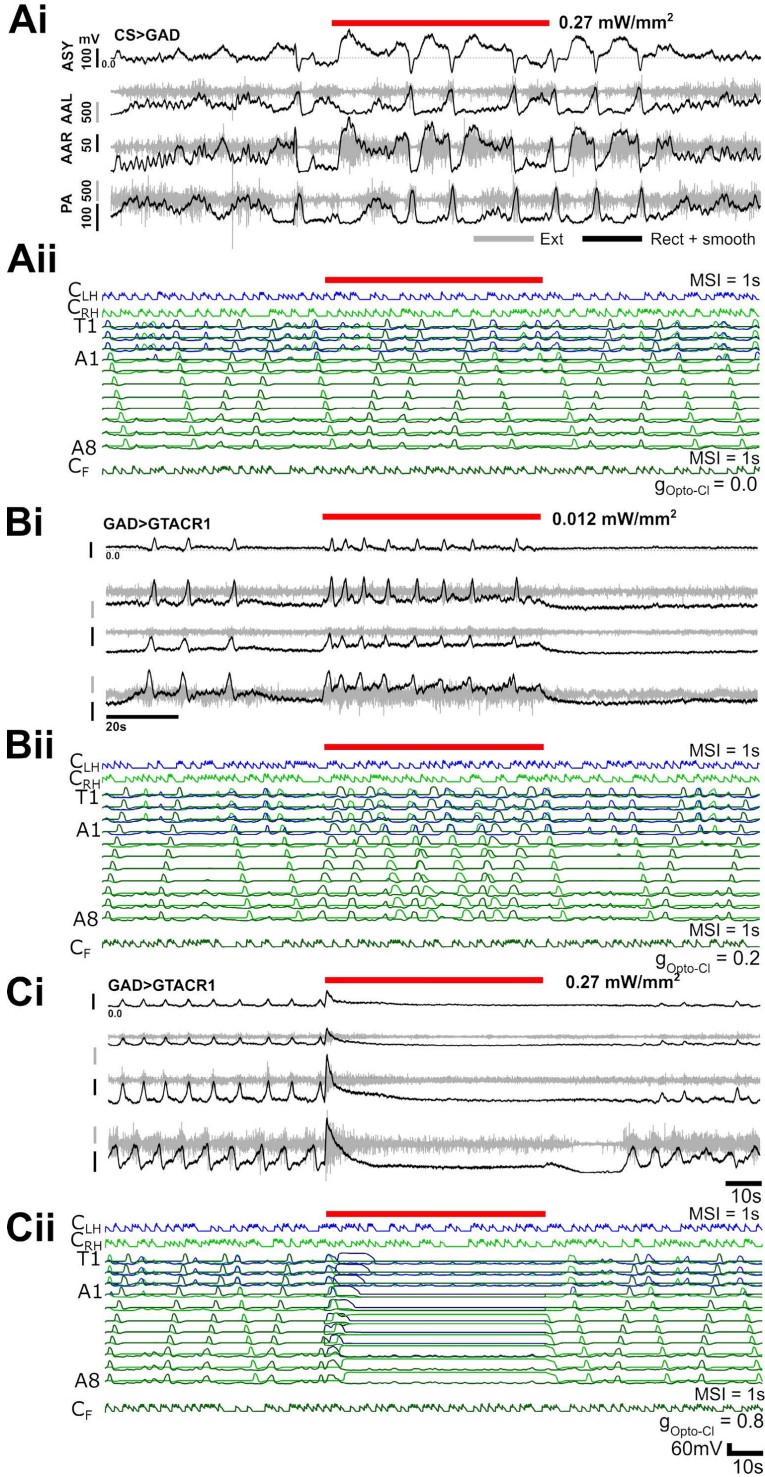

**Fig 14. Comparison of experimental CNS-wide optogenetic disinhibition and simulated disinhibition in the full CNS model.** In experimental preparations, the light gated anion channel GtACR1 is expressed in GABAergic neurons and motor output in response to red light pulses is recorded simultaneously from 3 motor nerve roots. Raw traces shown in grey, rectified, low pass filtered traces shown in black. Motor nerve root locations: PA = posterior abdomen, AAL = anterior abdomen left, AAR = anterior abdomen right, ASY = Asymmetry (computed trace). In simulations, a GtACR1 conductance is simulated in all inhibitory neurons with the same time course as in experimental preparations. See Methods for further details. **(A)** Heterozygous control preparation does not respond to light pulse (i). Similarly, simulation in which maximal GtACR1 conductance is set to 0 shows no response

(ii). **(B)** Low light levels trigger increased bursting in experimental preparations (i) and increased bursting and overlap of motor programs in simulations (ii). **(C)** At higher light intensity, bursting activity ceases in both experimental preparations (i) and simulations (ii). In A-C, traces are representative of results in n = 6 (experimental, GAD x GtCAR1), n = 5 (control, CS x GtCAR1), and n = 3 (control, CS x GAD) preparations. See S3 Fig for quantification. 10.17630/779141ce-c26a-483b-bfee-4f12cf71d7b2.

### Increasing levels of disinhibition generate overlaps in motor programs, followed by eventual collapse of rhythmogenesis

Next, we compared the effects of functionally removing inhibition in both experimental preparations and our computational model. Experimentally, we expressed the light-gated anion channel, GtACR1, in all GABAergic neurons using the GAL4-UAS system, and then recorded extracellularly from multiple motor nerve roots during pulses of varying light intensity. In the computational model, a simulated inducible anion channel was added to all inhibitory neurons (see Methods for details). Genetic controls did not respond to light pulses, and, not surprisingly, neither did computational models when the maximal conductance of the simulated anion channel was set to 0 (Fig 14A).

As light intensity was increased, experimental preparations showed an increase in bursting activity (Fig 14Bi), followed by complete collapse of rhythmic motor output (Figs 14Ci, S3). Similarly, in the model, a small increase in the conductance of the simulated anion channel, resulting in low levels of disinhibition, led to an increase in motor program overlap. This translated into increased frequency of bursting within any given hemisegment. As the strength of simulated disinhibition increased, rhythmogenic activity collapsed. All effects in experimental and computational experiments were reversible.

### Inhibitory motifs have specific roles in segregating motor programs and generating rhythms

To explore the dynamic roles that each inhibitory motif played within the full integrated model and to generate testable predictions for future experimental work, we examined how removing specific inhibitory motifs shapes the output of our full integrated model. Inhibiting activity in initiation detectors led primarily to overlap of the different motor programs as well as an increase in posterior bursting (Fig 15Ai,ii, n = 3 trials). In contrast, inhibiting activity in wave detector neurons led primarily to an increase in overlap amongst similar motor programs (Fig 15Bi,ii, n = 3 trials). Inhibiting inhibitory neurons within HO oscillators led to collapse of rhythmic activity (Fig 15Dii,ii, n = 3 trials). There was considerable variation in the distributions of activity patterns during control periods, but overall, these experiments were consistent with modelling experiments performed with abdominal circuits in isolation (Fig 7). Direct simulated current injection into inhibitory neurons produced effects similar to those produced by simulated optogenetics experiments. Slowly ramping up hyperpolarising current injection to all inhibitory neurons revealed dynamics of progressive disinhibition which was characterised first by overlap in opposing motor programs, followed by increased bursting and then eventual collapse of all rhythmic activity (Fig 15Ei, 15Eii). Excitatory drive from $E_F$ and $E_B$ neurons is able to overcome hyperpolarization of wave detectors, enabling segregation of the same motor programs right up until complete collapse of network activity.

## Discussion

We have developed an integrated network model with low-intensity command inputs and a set of relatively simple inhibitory motifs that can recapitulate the variability and diversity of motor programs exhibited in *Drosophila* larval CPG preparations. Inhibitory motifs with segments along the A-P axis generate oscillations locally, then feedforward excitation generates wave-like progression along the A-P axis. Intersegmental inhibitory motifs then enable detection of a given motor programme and suppression of competing motor programmes enhance the diversity and variability of motor programmes, while preventing maladaptive overlap of competing motor programmes. Diversity and variability in network output then emerges at a property of these tiered networks of inhibitory circuit motifs.

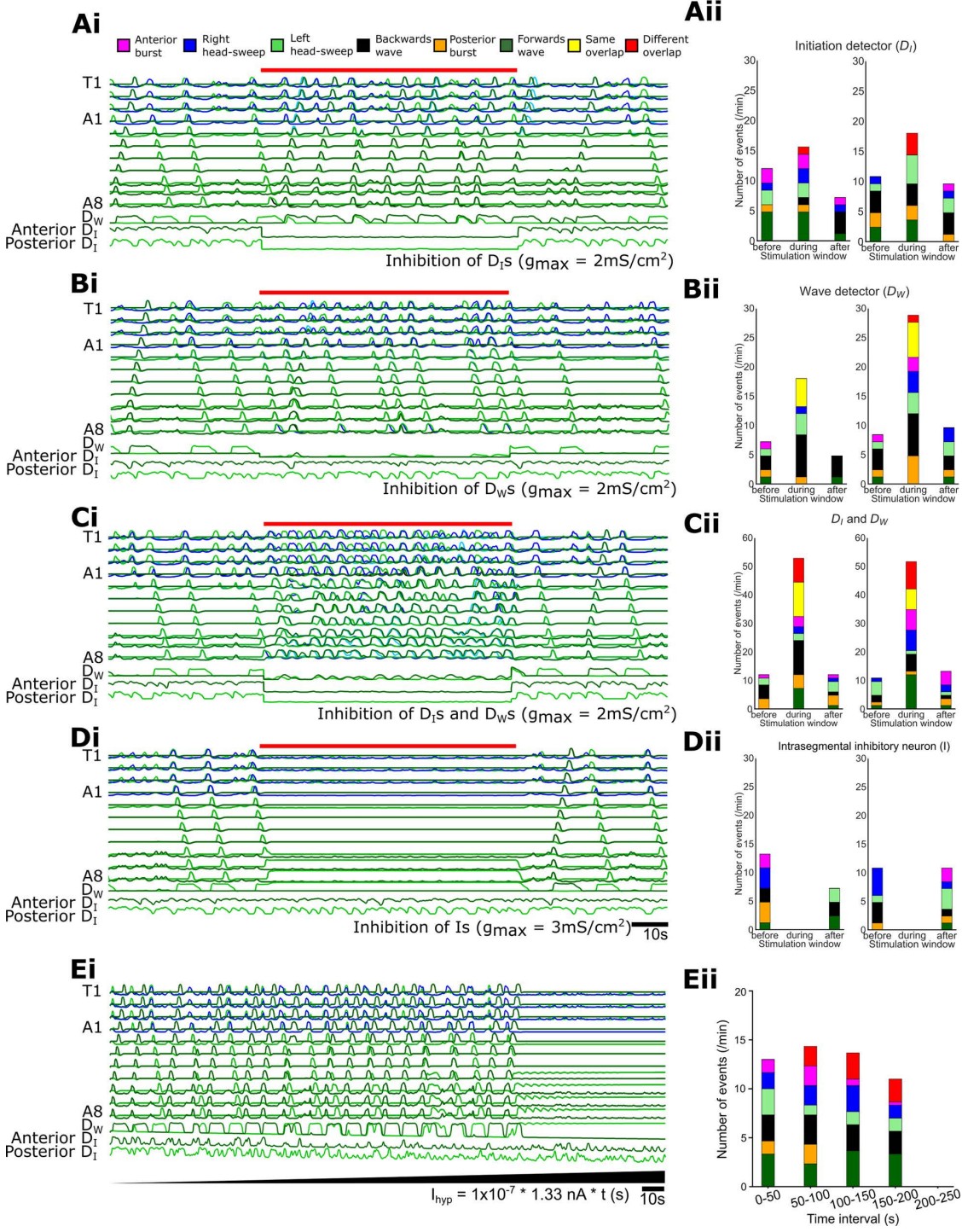

**Fig 15. Progressive removal of specific inhibitory motifs from model provides testable predictions for experimental work. (Ai)** Output of full network model before, during, and after simulated activation of GtACR1 in initiation detectors. **(Aii, Bii, Cii, Dii)** Proportion of different activity patterns in each condition (n = 2 simulations each). Note increase in overlap of same motor patterns. **(B, C, D)** Same as in (A) but with wave detectors alone, initiation detectors plus wave detectors together, and HO inhibitors alone inhibited, respectively. **(Ei)** Network output in response to slowly increasing direct hyperpolarization of all inhibitory neurons. **(Eii)** Proportion of different activity patterns at different time points during slow ramping disinhibition. Model results predict phenotypes associated with loss of functional motifs in experimental systems. 10.17630/779141ce-c26a-483b-bfee-4f12cf71d7b2.

## Modelling approach and scope

Here we present a circuit model of CPG networks that reproduces key features of motor program diversity and variability in the *Drosophila* larval locomotor system. The *Drosophila* community has generated a complete connectome of the larval brain and a nearly complete connectome of the larval ventral nerve cord [13]. Connectomics has provided deep insights into the functional architecture of circuits underlying larval behaviour, but does not, in itself, lead to an intuitive understanding of how rhythmic motor programs are generated in this system. To better understand the origins of CPG activity in our system, we created a circuit architecture that builds on simple components, first principles, and fundamental constraints, while being guided and inspired by available connectomics data. The point of our work was not to recapitulate all aspects of available connectomics data, but rather to synthesise ideas and concepts from anatomy, physiology and behavioural studies to study how diversity and variability in motor networks can emerge from a limited set of circuit components.

We see this work as a starting point, rather than an endpoint, directed towards a community conversation about how rhythmic motor programs are generated and coordinated in this and other model organisms. We welcome debate and discourse from experimentalists and computational neuroscientists alike, and so we have deliberately constructed our model using a powerful but easy-to-use modelling platform (Neurosim). The platform is scalable, low cost and requires no knowledge of computer programming. This makes it relatively easy for anyone to introduce additional complexity to the model, create new variants with a common foundation, and/or quickly test entirely new ideas. This approach is designed to lower energy barriers for the model to evolve and expand in scope organically as knowledge in the community grows.

## Why use simple non-spiking circuit elements?

We used non-spiking neurons and graded synapses in our model because it allowed cellular interactions to generate oscillations on long timescales (multiple seconds), similar to those observed in the biological system. It also allowed us to encapsulate and represent calcium imaging data and broadscale coarse connectivity patterns amongst oscillators and across motor programs. This type of synapse is also conceptually similar to the types of connections used in more abstract models [12,21]. Of course, we do not suggest that the animal uses exclusively non-spiking interactions; indeed, there is clear evidence from intracellular recordings that many interneurons in the larval locomotor network use spike-mediated neurotransmission (e.g., A27h; [31]). However, given that non-spiking interneurons and synapses are known to co-exist in the nervous system of many arthropods [39], it is reasonable to postulate their existence in our highly reduced model. Overall, the use of non-spiking single compartments and graded synapses here serves to connect our work to previous studies, highlight potential core circuit motifs, and also spark future exploration of these types of cells and synapses in the larval locomotor network. An interesting future avenue for modelling studies could be to explore how adding in spike-mediated synaptic transmission and multi-compartment models shape diversity and variability of the system's outputs.

## Hemisegmental oscillator

We chose excitation-inhibition (EI) interactions based on principles of Wilson-Cowan oscillators as the core mechanism for generating oscillations in our model. This allowed us to build on previously published work [21], and to incorporate motifs abstracted from identified neurons in the larval system, in addition to results from this experimental work and our previous studies [26]. Our experimental work with isolated posterior segments provides evidence for independent oscillators in hemisegments that require GABAergic transmission. Unlike other segmentally organised motor systems, we cannot excise single hemisegments for physiological analyses. But we reasoned that if small groups of hemisegments in posterior regions are able to slip in and out of phase with one another and run at different frequencies simultaneously, then they are unlikely to be driven by a common shared pacemaker. Furthermore, motor neuron bursting activity is

completely abolished when all GABAergic neurons are optogenetically hyperpolarized, and so it is likely that GABAergic neurons play a critical role in rhythmogenesis. This is complementary to previous work demonstrating that excitatory muscarinic signalling provides a functional basis for oscillations in reduced preparations [26]. Taken together, this body of work suggests that the basic unit of rhythm generation in our system is a conditional oscillator in each hemisegment which generates rhythms based on balances of excitation and inhibition mediated by local muscarinic and GABAergic signalling. This is consistent with previous work that has shown the critical importance of muscarinic signalling and inhibition for rhythm generation in other arthropods [40–43]. The role of inhibition in vertebrate spinal networks is less clear-cut, with inhibition playing diverse roles [44–46] reviewed in [47], but appearing to actually be dispensable for some types of rhythmic activity [48].

Our HO conditional oscillator model is consistent with experimental results in our system, but oscillators with similar output properties could conceivably be constructed based on alternative mechanisms such as half-centre mutual inhibition [48,49], endogenous burster neurons [50], or even long-lasting post-burst hyperpolarisations generated by the sodium/potassium pump [23,51]. Indeed, it is possible that oscillations are a result of hybrid mechanisms incorporating several such effects [52]. One way to begin to parse out these possibilities would be to test the prediction that each larval hemisegment contains an HO oscillator motif in which a GABAergic neuron or neurons have local reciprocal connections with separate populations of excitatory neurons that are active in time with or before motor neurons within a hemisegment, and are recruited selectively into forwards and backwards waves. We would expect that some or all of these neurons would have muscarinic receptors, and that the excitatory populations would have mutual excitatory connections with sibling neurons across the midline in abdominal, but not thoracic segments. Optogenetically inhibiting one of the excitatory populations should not affect rhythm generation in the inhibitory neuron and opposing excitatory population, but inhibiting activity in the inhibitory population should collapse all rhythmic activity. If our HO model is accurate, we would also expect that none of the core HO neurons would be intrinsically bursting when synaptically isolated from each other in the presence of pharmacological blockers for synaptic transmission. Future experimental studies could test these predictions with existing tools and approaches.

## Mechanisms for generating irregularity

A striking feature of the isolated larval CNS is its ability to produce a wide variety of motor programs with varying frequencies. Recapitulating the highly diverse and variable CPG output observed in experimental preparations in a computational setting represents a challenge, but is also a fundamental constraint for any model of larval locomotor CPGs. Previous models of larval locomotor CPGs produced highly regular forwards and backwards wave-like activity with consistent intersegmental phase relationships across wave speeds [21]. Our work incorporates stochastic inputs from command neurons which trigger conditional HOs and segmental motor programs. This type of input could conceivably be generated and/or modulated by sensory inputs or neuromodulators in the animal, resulting in biases toward certain types of motor programs. Additional Inhibitory motifs that detect wave-like activity in one motor programme and then suppress initiation and/or execution of opposing motor pogrammes allow for a diverse array of network activity while also providing a backstop that prevents maladaptive overlap of behaviours. This type of organisation, with segmentally coupled conditional HOs as core rhythm generating units, provides one explanation for how irregular, varied activity in larval CPGs could be generated and then transformed into more regular, less varied behaviour in the intact animal [17].

Irregularity is often equated with poor health in a CPG network, but in our preparations, it appears rather to be the natural state of the CPG network in the absence of sensory feedback and a body. Similar degrees of irregularity have been seen in other CPG networks that coordinate movement of soft bodies over terrain [53]. In limbed animals with internal or external skeletons, CPG activity is also often very irregular, and only becomes regularised in the presence of pharmacological agents [54,55] or during optogenetic activation of neuromodulatory neurons [56].

## Integration with experimental work on larval CNS circuits

There are conceptual similarities in motifs, but also some differences in the architecture of our model compared to previous experimental work. The basis of the intersegmental connections between hemisegmental oscillators (HO) reflects the relationship between inhibitory 'GDL' and excitatory 'A27h' neurons revealed by [31]. In our model, the forwards excitors, backwards excitors and hemisegmental inhibitors conceptually resemble information encoded in A27h and A18b premotor interneurons and GDLs in that inhibitor neurons feed inhibition to excitatory neurons within a segment and excitors feed excitation within and to neighbouring segments. In the model, excitatory pathways from forwards and backwards excitors in each hemisegment converge onto wave detectors. This conceptually resembles the information encoded by populations of ascending cholinergic neurons such as 'canon' [35] and 19f/M [36]. In the animal, the cell bodies of these neurons reside in each hemisegment and they are strongly interconnected with each other, and premotor interneurons. As would be expected, both cell types are recruited into motor programs in the isolated CNS, and both have projections that converge on inhibitory interneurons. Manipulating activity in these neurons appears to suppress motor activity and in some cases, modulate segregation of motor programs. In the model, this pathway is compressed into a simple set of excitatory synapses from excitors onto a wave detector; however, in the animal, there is an additional layer of integration which could serve to enrich and modulate information sent to wave detectors about the state of ongoing motor programs.

An additional divergence between the model and reported connectomics data revolves around the MDN-Pair 1 axis. In our model, the command head-sweep ($C_H$) population resembles MDN neurons in that they excite an intermediary neuron ($D_{IB}$) – akin to Pair 1 – which inhibits the posterior $E_f$. However, the target site of $C_H$ in our model is not A1 but the neighbouring thoracic regions of the VNC network and, unlike MDN neurons, $C_H$ drives anterior asymmetric activity. This promotes fictive backwards activity by driving activity in backwards promoting excitatory neurons ($E_B$), but slightly more anterior than in published work. Furthermore, Pair 1 neurons appear to be recruited during both backwards waves and command initiation, suggesting that in the animal, they may have a combined role as both wave and initiation detectors.

Our work points towards the importance of wave detector motifs for the segregation of same motor programs and initiation detector motifs for the segregation of different motor programs. One testable prediction from our work is that these specific roles and corresponding anatomical motifs will be reflected in populations of inhibitory interneurons. We also predict that progressive network-wide disinhibition will lead first to overlap in opposing motor programs, followed by a progressive increase in rhythmic activity without overlap of same motor programs, and end with complete collapse of all rhythms. Examining recruitment patterns of inhibitory interneurons and overlap between same and different motor programs when sets of inhibitory interneurons are silenced would allow experimenters to confirm or refute these predictions.

Our model aimed to replicate the diversity and complexity of motor programs in an isolated CNS with a small number of simple elements. Other circuit features, while not built into this version of the model, could be integrated to examine how larval behaviours are generated and controlled. For instance, circuits that work to generate phase delays amongst antagonistic muscle groups could be added to create intrasegmental [57] and intersegmental [34] patterning, as well as circuits that control locomotion speed [58]. Furthermore, the head-sweep command neurons ($C_{HL/R}$) motif could be modified to encode or reflect olfactory gradients as opposed to the Poisson statistics this model is built upon; specifically, the model could be adapted to explore the role of PDM-DN neurons that integrate olfactory gradients into inhibition of forwards peristalsis through descending SEZ-DN1 neurons [59] or the integration between Goro command-like neurons and MDN in rolling behaviour [60]. Recent findings in adult flies by Braun and colleagues, [61] detail the hierarchical recruitment of command-like descending and population descending neurons; similar motifs could also be introduced upstream to command neurons in our model. Finally activity from stretch receptors and other sensory neurons embedded in the larval body wall could be layered into the model to explore how sensory input from specific cell types sculpts the activity HOs, detector motifs and command inputs [62–65].

## Regulation and coordination of motor program diversity

Many motor systems are capable of producing diverse motor patterns. The regulation and coordination of diverse patterns over time can take different forms, including 1) the simultaneous expression of and entrainment between several rhythmic activity patterns, such as the pyloric and gastric rhythms in the crustacean stomatogastric system [66], or fictive walking and flight in deafferented locust preparations [43]; 2) switching between different activity patterns at regular time intervals, as in the leech heartbeat system [67,68]; 3) switching between hierarchically organised motor behaviours in which a (often escape-related) motor program such as a tail flip [69], wing withdrawal [70], struggling behaviour [71], or escape rolling [72] interrupts and suppresses an ongoing motor pattern; and finally 4) flexible and adaptive switching between diverse motor patterns to meet moment-to-moment behavioural needs, such as those encountered when an animal moves through and must respond to a complex environment.

Our larval CPG circuit model is intended to form the core of a circuit model that falls into categories (3) and (4). As such, it must be able to produce different activity patterns (forwards and backwards waves, left and right head sweeps) that are mutually exclusive, and ensure that each can be fully and reliably executed once initiated. Furthermore, the circuit must allow easy switching from one pattern to another on a short timescale, and biasing of the circuit output toward one motor program or another on a longer timescale, to allow for adaptive pattern adjustments in response to changing behavioural requirements.

Surprisingly, we have found that many of these requirements can be achieved on the basis of relatively simple inhibitory motifs and command-like circuit components such as $C_{HL}$, $C_{HR}$, and $C_{FL/R}$. In the absence of explicitly modelled sensory feedback to the CPG network or behavioural state signals, the stochastic command signals driving the model generate diverse activity patterns that resemble those seen in the biological network. Our results show that adjustments to the MSI ratios of the command-like signals can bias network output toward one activity pattern over another. In future work, we envision embedding this CPG circuit model in an extended CNS model in which input to the command-like neurons depends on simulated sensory inputs. For example, traversing a nutrient concentration gradient could be mimicked in an extended model by providing biased command inputs that favour head sweeps and crawling up the gradient. On a shorter timescale, periodic inputs to the appropriate command neurons in the circuit model could be used to cause head sweeps followed by movement in a different virtual direction in response to bumping into simulated obstacles.

Our work does not necessarily preclude the presence of other mechanisms for generating motor program diversity in the larval CNS. Indeed, circuits for head-sweeps and wave generation are connected to other circuits controlling other behaviours such as feeding [73], hunching (head retraction and simultaneous contraction of body wall muscles) [1], and rolling [74], and it is possible that some of the diversity we see is due to simultaneous expression of other motor programs along the lines of category (1). To date, we have not seen evidence that there is any type of 'internal clock' that consistently triggers switches in motor programs within isolated CNS preparations; however, previous work has shown evidence for a highly rhythmic motor program underlying head-sweeps present in the background of larval locomotor behaviours which is amplitude modulated [20]. Further, previous work has clearly shown that activation of specific interneurons and/or sensory neurons can override ongoing programs and trigger escape-like behaviours in larvae [60,63]. Overall, switching among forwards, backwards, head-sweep and other motor programs in intact animals appears not to be rigidly imposed, but rather highly dependent on integrating information from sensory landscapes.

## Diverse and variable outputs for CPG networks controlling movement over terrain

Compared to aquatic and aerial locomotion, terrestrial locomotion often occurs in a physically less homogenous environment. Terrestrial locomotor rhythm- and pattern-generator networks thus must be able to adjust the motor patterns they produce on the step or crawl cycle-by-cycle timescale (i.e., adjusting step length to terrain and obstacles), and on the timescale of multiple cycles within the same motor program (i.e., switching gait). A circuit that produces highly regular and

stable motor patterns and "locks into" a stereotyped activity mode for prolonged periods is not the best answer to this challenge. Rather, flexibility and ease of switching between different CPG activity patterns is required. Our results demonstrate that inhibitory motifs that prevent simultaneous activation of conflicting motor programs can allow for such flexibility.

While a full scale dynamical systems analysis of our CPG circuit model is beyond the scope of this paper, our circuit model appears to be situated near the boundaries between diverse activity programs, rather than being firmly entrenched in a single highly stable and stereo-typed activity pattern. This is also reflected in the variability seen in activity features such as, for example, the period of consecutive activity cycles. Diversity and variability is higher in our circuit model and in isolated preparations compared to the intact animal [17], suggesting that diversity and variability indeed arises from CPG circuits themselves, and in the intact animal are modulated by sensory inputs and constrained by the biomechanics of the body. In this sense, motor pattern diversity and variability are beneficial consequences of CPG circuit architecture, rather than design flaws that must be overcome, or nuisances that must be minimised or averaged out by the researcher.

**Table 1. Neuron properties. The membrane parameters are specified per unit area (cm-2), and scaled to a spherical diameter.**

| *General* | |
| --- | --- |
| Temperature (for Nernst calculation) | 294K |
| Extracellular calcium concentration | 2 mM |
| *Neuron* | |
| *Passive* | |
| Diameter | 10 $\mu m$ |
| Membrane capacitance | 1 $\mu F$ cm$^{-2}$ |
| Leak conductance (fixed) | 0.35 mS cm$^{-2}$ |
| Leak equilibrium potential | −60 mV |
| *Voltage-dependent calcium channel* | |
| Maximum conductance ($g_{max}$) | 0.08 mS cm$^{-2}$ |
| Mid-activation voltage ($V_{mid}$) | −40 mV |
| Sensitivity ($S_{Ca}$) | 7.5 mV |
| Activation time constant ($\tau_m$) | 50 ms |
| *Intracellular calcium concentration* | |
| Inflow factor ($B$) | 1,000 |
| Clearance time constant ($\tau_{Ca\_clr}$) | 400 ms |
| Minimum concentration ($Ca_{i\_min}$) | 0.1 nM |

**Table 2. Synapse properties.**

| | excitatory | inhibitory |
| --- | --- | --- |
| Equilibrium potential (mV) *x_eq* | −10 | −70 |
| Maximum release rate (ms$^{-1}$) $r_{max}$ (bilateral coordination) | 80 20 | 80 |
| Mid calcium concentration (nM) $Ca_{i\_mid}$ | 185 | 200 |
| Calcium release sensitivity (nM) $S_r$ | 15 | |
| Transmitter clearance time constant (ms) $\tau_{T\_clr}$ | 250 | |
| Receptor binding sensitivity $S_{bind}$ | 0.5 | |
| Maximum conductance (mS cm$^{-2}$) $g_{syn\_max}$ | 1 | |
| Gain modulator (inhibitory initiation and wave detector output) | 1 2 | |

## Materials and methods

### Model implementation

The computational model was built using Neurosim 5 [75]: https://www.st-andrews.ac.uk/~wjh/neurosim). This neural simulator allows modelling at the cellular and small systems level, and it has an intuitive user interface that facilitates the rapid development, modification, and testing of simulated neural circuits. Binary Neurosim parameter files that can produce the outputs described in this paper are freely available upon request and also available within the data repositories for this publication.

In Neurosim, equations are solved using the exponential Euler integration method [76], and we specified a fixed step size of 1 ms. Default parameter values for the various equations listed below are given in the text and in Tables 1 and 2. All these values are exposed and modifiable in the user interface within Neurosim, as is the overall circuit configuration. When parameter values were altered as part of an experimental procedure, this is noted in the Results text.

### Neurons

Neurons were simulated as single-compartment models implementing a Hodgkin-Huxley type formalism. The neurons do not generate spikes, but have non-inactivating voltage-dependent calcium channels that allow them to develop plateau-like potentials on receiving a depolarizing stimulus. Synaptic transmitter release is graded, and is controlled by the instantaneous intracellular calcium concentration. All neurons in the circuit are physiologically identical with a spherical diameter of 10 $\mu$m.

Each neuron contains three types of ionic channels: leakage, voltage-dependent calcium, and synaptic (the latter with various subtypes), each with an instantaneous conductance $g_x$. For leakage channels $g_x$ is fixed (0.35 mS cm$^{-2}$), for the other channels $g_x$ varies up to a maximum $g_{x\_max}$ ($g_{Ca\_max}$ = 0.08 mS cm$^{-2}$, $g_{syn\_max}$ = 1 mS cm$^{-2}$) dependent on a controlling factor $m_x$, which varies between 0 and 1.

$$g_x = m_x g_{x\_max} \tag{1}$$

The channels are assumed to be Ohmic, so the current ($I_x$) flowing through each channel is:

$$I_x = g_x (V_m - V_{x\_eq}) \tag{2}$$

where $V_m$ is the membrane potential, and $V_{x\_eq}$ is the equilibrium potential of the ion(s) for which the channel is permeable. Leakage and synaptic channels have fixed equilibrium potentials ($V_{leak\_eq}$ = −60 mV, $V_{syn\_eq}$ = −10 mV for excitatory synapses, −70 mV for inhibitory synapses), but the calcium equilibrium potential is updated with changes in the intracellular calcium concentration ([$Ca_i$], initial value = 20 nM) during the course of a simulation, according to the Nernst equation (extracellular calcium concentration [$Ca_e$] = 2 mM, temperature = 294 K).

The membrane potential of each neuron is calculated by integrating the current-balance equation:

$$C_m \frac{dV_m}{dt} + \sum I_x - I_{stim} = 0 \tag{3}$$

where $C_m$ is the membrane capacitance (1 $\mu$F cm$^{-2}$), $I_x$ is the ionic current through channel type x (equation 2) and $I_{stim}$ is current from an external stimulus.

***Voltage-dependent calcium channel.*** Voltage-dependent calcium channels contain a single activation gate whose open probability constitutes the controlling factor $m$. The steady-state value of $m$ ($m_\infty$) is voltage dependent according to the sigmoid (standard logistic) function.

$$m_\infty = \frac{1}{1 + e^{(V_{mid} - V_m)/S_{Ca}}} \tag{4}$$

where $V_{mid}$ is the mid-activation voltage (−40 mV) and $S_{Ca}$ is the sensitivity of the slope (7.5 mV). The value of $m$ is updated by integrating:

$$\frac{dm}{dt} = \frac{m_\infty - m}{\tau_m}$$

(5)

where $\tau_m$ is a fixed-value time constant (50 ms), and the initial value is $m_\infty$.

**Intracellular calcium concentration.** Calcium enters the cell as current ($I_{Ca}$) through voltage-dependent calcium channels, but only distributes within a restricted fraction of the cell volume. A calcium inflow factor ($B = 1,000$ mM s$^{-1}$nA$^{-1}$) converts current to rate of concentration change within this volume fraction. Calcium is cleared from the cell to a minimum level ($Ca_{i\_min} = 0.1$ nM) at a concentration-dependent rate with a specified time constant ($\tau_{Ca\_clr} = 400$ ms). Intracellular calcium concentration $Ca_i$ is updated by integrating the difference between the rate of inflow and the rate of clearance:

$$\frac{dCa_i}{dt} = B\,I_{Ca} - \frac{Ca_i - Ca_{i\_min}}{\tau_{Ca\_clr}}$$

(6)

## Synapses

Post-synaptic conductance depends upon the nominal transmitter concentration ($T$: unspecified unit scale) in the synaptic cleft. Transmitter is released into the cleft at a rate that is a sigmoid function of the pre-synaptic intracellular calcium concentration, with a maximum rate of $r_{max}$, (20 for excitatory synapses mediating bilateral coordination in abdominal segments, 80 for all other synapses), a mid rate at concentration $Ca_{i\_mid}$ (excitatory: 200 nM; inhibitory: 185 nM) and a slope sensitivity of $S_r$ (15 nM). Transmitter is cleared from the cleft at a concentration-dependent rate with a fixed time constant ($\tau_{T\_clr}$: 250 ms). Cleft transmitter concentration is updated by integrating the difference between between the rate of release and the rate of clearance:

$$\frac{dT}{dt} = \frac{r_{max}}{1 + e^{(Ca_{i\_mid} - Ca_i)/S_r}} - \frac{T}{\tau_{T\_clr}}$$

(7)

The post-synaptic conductance control factor $m_{syn}$ is an upper-half sigmoid function of the nominal transmitter concentration in the cleft,

$$m_{syn} = \frac{2}{1 + e^{T/S_{bind}}} - 1$$

(8)

where $S_{bind}$ (0.5) is the sensitivity of the slope that characterises binding of the transmitter to the post-synaptic receptor.

**Connection gain modulation.** Each synaptic connection has a gain modulator that multiplies the postsynaptic conductance after it has been calculated as described above. Hence a modulator value of 1 has no effect, a value greater than 1 increases the synapse strength of that particular connection, a value less than 1 decreases the strength. All connections have a gain of 1, except the inhibitory output connections of the wave detectors ($D_{WF, WB}$, $D_{IF, IB}$) which have a gain of 2.

## Optogenetic disinhibition

To simulate optogenetic disinhibition an additional channel was placed in all inhibitory neurons with an equilibrium potential of −70 mV to simulate chloride specificity. The channel had a fixed conductance ($g_{opto\_Cl} = 0.2$ or 0.8 mS cm$^{-2}$) but was normally completely blocked. To simulate the effect of illumination the channel was unblocked at the appropriate moment in a simulation run. This caused strong inhibition of the inhibitory neurons, and thus disinhibited the circuit as a whole.

## Animal rearing and genetic constructs

*Drosophila melanogaster* larvae were reared in vials using standard cornmeal-based food in vials. In imaging experiments, flies were genetically modified using the GAL4-UAS system to drive expression of the calcium indicator GCaMP6m [77] in all glutamatergic neurons, including the motor neurons, through OK371-GAL4 [78]. All imaging and perfusion experiments used OK371-GCaMP6m 3rd instar larvae. For optogenetic and electrophysiology experiments, a GAD1-GAL4 driver generated using a 'Trojan Exon' approach [79] was used to drive expression of the green and red-light gated anion channel GtACR1 [80]. All animals were reared at 23–25°C with an approximately 12h:12h light-dark cycle.

## Isolated CNS dissection

Individual 3rd instar larvae were positioned and pinned on Sylgard-lined petri dishes dorsally through the mouthparts and posterior abdomen. Using fine scissors, an incision along the dorsal surface of the body wall enabled removal of all internal organs. The body wall was pinned flat exposing the CNS. The brain, suboesophageal ganglion (SOG), and ventral nerve cord (VNC) were dissected and pinned using fin tungsten wire (California Fine Wire, Grover Beach, CA) with remaining tissue removed before imaging. The dish was washed five times with Baines External Saline (BES). Before imaging, fresh BES was applied containing (in mM): 135 NaCl, 5 KCL, 2CaCl$_2$, 4 MgCl$_2$, 5 TES and 36 Sucrose. All imaging recordings were made within 15 min after ablation.

## Live calcium imaging in isolated VNC

Live preparations were imaged using an Olympus UPlanFL 10x INFINITY-corrected camera with 0.3NA (UPLFLN10X2). Images were captured using WinFluor (4.1.9) at 10fps. During imaging, all preparations were perfused with BES for up to 45 min. Images were stabilised against lateral shifts using a Template Matching plugin [81] in FIJI [82]. Fluorescence values were manually extracted using regions of interest (ROIs) per thoracic (T1-3) and abdominal (A1-A8) segment within FIJI as reported in [17]. The timeseries of extracted fluorescence values were extracted and pre-processed using a custom python script creating percentage change in fluorescence from baseline values (ΔF/F) before being visualised using DataView.

## Electrophysiology and optogenetics

Live imaging was performed on a customised Olympus BX51wi fluorescence microscope with an Andor iXon camera (Andor Technology, Belfast, UK) which was triggered from a National Instruments Data Acquisition (NIDAQ) board 6,229 (National Instruments, Austin, Texas, US). Winfluor v4.0.3 (University of Strathclyde, Glasgow, UK) was used to send trigger pulses and images were acquired at 10 Hz. Suction electrodes were mechanically moved toward VNC using an MP-285 mechanical micromanipulator (Sutter Instruments, Hitchin, UK) or moved manually using a hand-operated manipulator. Electrodes were attached onto different motor nerve roots protruding from the VNC. The electrodes were connected to an A-M Systems Model 1,700 extracellular amplifier (A-M Systems, Sequim, WA, USA), low pass filtered at 300 Hz, and high pass filtered at 500–1,000 Hz. The output from the amplifier was input into a PowerLab acquisition board (AD Instruments, Dunedin, New Zealand). Samples were activated with an OptoLED lightsource (Cairn Instruments, Kent, United Kingdom) which was used to generate red 625nm LED pulses (light intensity ranged from 5μW/mm$^2$ to 3.62mW/mm$^2$ which was directed to the preparation using a dichroic mirror placed in a custom-built housing (Cairn Instruments). Red light stimulations were controlled using LabChart7 software (AD Instruments, Dunedin, New Zealand) (60 seconds). Comments were added during the LabChart recording to denote the type of stimulation performed.

## DataView and python analysis

Fluorescence data was visualised using DataView (11.17.1) [83]. Peaks in segmental activity were determined using the built-in hill-valley analysis facility. Bilateral T3 activity (i.e., fictive head-sweep) was determined using > 5% peak difference

between left and right fluorescence values. Instantaneous frequency was determined using in-built DataView functions. Fictive forwards and backwards activity patterns were defined as progressive peaks in ascending (A8→T3) or descending segments (T3→A8), respectively. Anterior and posterior burst activity was defined as near-synchronous peak times in anterior (T3 – A1) and posterior segments (A6 – A8), respectively (as reported in [17]). Fictive events were determined to be overlapping if a signal in the calcium signal at the fictive event's initiating segment (i.e., T3 for anterior burst, asymmetry and backward waves; A8 for posterior burst, and forward waves) was within a 20% up to 20% down event threshold from the central peak in each segment; an anterior event overlaps a posterior event if if both calcium signals are within 30% of the peaks in calcium activity in each segment. (compare Fig 7Bii as same-overlap and 7Diii as different-overlap events). In the computational model, a same-overlap event was recorded when two events originating from the same region, namely, either forwards or backwards waves, resulted in propagation of two concurrent waves in the same metachronal direction. Different-overlap events were defined as events where a collision at any abdominal segment between forwards and backwards waves occurred (compare Fig 7Bii as same-overlap and 7Diii as different-overlap events). Intersegmental delay was calculated for peak time in each segment relative to A8 and T3 for fictive forwards and backwards activity, respectively. The kernel density estimate (KDE) for bilateral T3 activity was computed through Seaborn using a Gaussian kernel and a bandwidth multiplicative factor of 4 [84]. Transition matrices were calculated by quantifying the percentage of subsequent motor programs that followed each motor program.

## Supporting information

**S1 Fig. Kernel Density Plots for all 15 isolated VNC calcium imaged preparations used in model comparisons.** Kernel density plots showing all preparations that exhibited a fictive right (A), no (B), and left (C) asymmetric bias. 10.17630/779141ce-c26a-483b-bfee-4f12cf71d7b2. (TIF)

**S2 Fig. Probability transition matrices for all 15 isolated VNC calcium imaged preparations.** 10.17630/779141ce-c26a-483b-bfee-4f12cf71d7b2. (TIF)

**S3 Fig. Quantification of Anterior and Posterior Activity and Overlap in Disinhibition Preparations.** (A) Count per minute activity in anterior (T3) and posterior (A4-6) segments in GAD x GtCAR1 and control CS x GtCAR1 preparations. (B) Percentage of overlap between the anterior and posterior activity during the 60s stimulation period. Note, disinhibition at low optogenetic stimulation (green, $0.012\,\mu W/cm^2$) induces increased activity rate with more overlap between anterior and posterior segments whereas disinhibition at higher optogenetic stimulation (blue, $0.27\,\mu W/cm^2$) collapses rhythmic activity across segments. 10.17630/779141ce-c26a-483b-bfee-4f12cf71d7b2. (TIF)

## Acknowledgments

We thank Dr. Maarten Zwart for comments on an earlier version of this manuscript.

## Author contributions

**Conceptualization:** Jacob Francis, Caius R. Gibeily, William V Smith, Isabel S. Petropoulos, William J. Heitler, Astrid A. Prinz, Stefan R. Pulver.

**Data curation:** Jacob Francis, Caius R. Gibeily, William V. Smith, Isabel S. Petropoulos, Michael Anderson.

**Formal analysis:** Jacob Francis, Caius R. Gibeily, William V. Smith, Isabel S. Petropoulos, Michael Anderson, William J. Heitler, Astrid A. Prinz.

**Funding acquisition:** Jacob Francis, Caius R. Gibeily, William V. Smith, Astrid A. Prinz, Stefan R. Pulver.

**Investigation:** Jacob Francis, Caius R. Gibeily, William V Smith, Michael Anderson, William J. Heitler, Astrid A. Prinz, Stefan R. Pulver.

**Methodology:** Jacob Francis, Caius R. Gibeily, William V. Smith, Isabel S. Petropoulos, Michael Anderson, William J. Heitler, Astrid A. Prinz.

**Project administration:** Astrid A. Prinz, Stefan R. Pulver.

**Resources:** Jacob Francis, Caius R. Gibeily, William V. Smith, William J. Heitler, Astrid A. Prinz.

**Software:** Jacob Francis, Caius R. Gibeily, William V. Smith, William J. Heitler.

**Supervision:** Astrid A. Prinz, Stefan R. Pulver.

**Validation:** William J. Heitler, Stefan R. Pulver.

**Visualization:** Jacob Francis, Caius R. Gibeily, William V. Smith, Michael Anderson.

**Writing – original draft:** Caius R. Gibeily, William V. Smith, William J. Heitler, Astrid A. Prinz, Stefan R. Pulver.

**Writing – review & editing:** Jacob Francis, Caius R. Gibeily, William V. Smith, Isabel S. Petropoulos, Michael Anderson, William J. Heitler, Astrid A. Prinz, Stefan R. Pulver.

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
