## [Editor Report · Decision Letter 0]

9 Sep 2024

Dear Dr Pulver,

Thank you for submitting your manuscript entitled "Generation of motor program diversity and variability through inhibitory circuit motifs in the Drosophila larval locomotor system" for consideration as a Research Article by PLOS Biology.

Your manuscript has now been evaluated by the PLOS Biology editorial staff as well as by an academic editor with relevant expertise and I am writing to let you know that we would like to send your submission out for external peer review.

Once your full submission is complete, your paper will undergo a series of checks in preparation for peer review. After your manuscript has passed the checks it will be sent out for review. To provide the metadata for your submission, please Login to Editorial Manager (https://www.editorialmanager.com/pbiology) within two working days, i.e. by Sep 11 2024 11:59PM.

Kind regards,

Christian

Christian Schnell, PhD

Senior Editor

PLOS Biology

cschnell@plos.org

---

## [Decision Letter · Decision Letter 1]

1 Nov 2024

Dear Dr Pulver,

My name is Luke Smith - I am end editor at PLOS Biology and am writing on behalf of my colleague, Christian Schnell, who is away on vacation at the moment. Thank you for your patience while your manuscript "Generation of motor program diversity and variability through inhibitory circuit motifs in the Drosophila larval locomotor system" was peer-reviewed at PLOS Biology. Your study has now been evaluated by the PLOS Biology editors, an Academic Editor with relevant expertise, and by several independent reviewers.

In light of the reviews, which you will find at the end of this email, we would like to invite you to revise the work to thoroughly address the reviewers' reports.

As you will see below, the reviewers have highlighted that the study is generally well done and that it will be of high interest to the field. However each reviewer has provided suggestions to strengthen the study further and we think these should be carefully and thoroughly addressed. Reviewer 3 has raised, perhaps, the most substantial serious concerns which we think will be important to address in the revision, although we think in many cases, these concerns may be addressable through textual changes and providing additional discussion and clarifications.

Given the extent of revision needed, we cannot make a decision about publication until we have seen the revised manuscript and your response to the reviewers' comments. Your revised manuscript is likely to be sent for further evaluation by all or a subset of the reviewers.

**IMPORTANT - SUBMITTING YOUR REVISION**

*Re-submission Checklist*

*Published Peer Review*

*PLOS Data Policy*

*Blot and Gel Data Policy*

Sincerely,

Luke

Lucas Smith, PhD

Senior Editor

PLOS Biology

lsmith@plos.org

-on behalf of-

Christian Schnell, PhD,

Senior Editor

PLOS Biology

cschnell@plos.org

REVIEWS:

Reviewer #1: Generation of motor program diversity and variability through inhibitory circuit motifs in the Drosophila larval locomotor system

Francis et al. submitted to PLoS Biology.

The authors generate a computational model that can recapitulate several features of the neural activity observed in isolated nerve cords of the Drosophila larval CNS.

The first place where this paper departs from earlier work (e.g., Pulver et al., Jonaitis et al. 2022) is a much more detailed examination of fictive motor patterns (Figures 1-5) using calcium imaging in CNSs isolated from Drosophila larvae. These data confirm and extend prior observations, with, in my opinion, an interesting and important new treatment of "anterior bursts" and "posterior bursts" and a more nuanced understanding of axial diversity in wave propagation. These descriptions later go on to inform many of the modeling decisions. A major strength of this paper is the data-informed approach.

Similar to previous models (Gjorgjieva et al., 2013), this paper's model comprises chains of local excitatory and inhibitory oscillators that can propagate wave-like activity along the axis. Here, the details of the oscillators differ slightly, and likely importantly, from prior work. However, this is not fully discussed. More novel is the incorporation of a second layer of "surrounding architecture," including neurons that detect the initiation of a wave and neurons activated for the wave's duration.

This work is extremely significant to the field of Drosophila larval behavior. First, it summarizes a diversity of work into a simple, powerful, plausible model. Second, this summary is useful for predicting the logical architecture of locomotor circuitry. More broadly, this work will interest any reader interested in the neural basis of behavior and those interested in computational modeling, physiology, and connectomics.

The claims are properly placed in the context of previous literature. One small note is that if they cite Vaddia et al., 2019, they should also cite He et al., 2019, as those papers came out back to back in the same journal.

The data and analyses support the major claims. In my opinion, no further work is needed. One note is that several graphs' text and other visual elements are too small to read without additional magnification.

I found this paper to be well-written and enjoyable to read.

Reviewer #2: In this study, Francis et al. build a circuit-based model capable of recapitulating the diversity and variability of motor programs observed in isolated CNS preparations of Drosophila larva.

The authors first generate experimental data (Calcium imaging of fictive motor programs in isolated CNS) that they use, along with data from previously published studies to analyze the patterns of motor activity in these preparations. They characterize the type of motor programs they generate, their frequency as well as diversity within and between preparations.

They further build a model capable of generating rhythmic activity. The model is based of incorporating principles of circuit organisation in the larval motor system (based on published connectomics data) without including the full connectivity matrix of such a network. They build a minimal circuit module composed of two excitatory and an inhibitory neuron that is repeated across segmentz and incorporate elements (like for example wave detectors and command neuron) that allow to reproduce the features of the motor pattern activity observed in the larval CNS. They specifically explore the role of inhibitory motifs in generating oscillations and show (both experimentally and in the model) that inhibition in the circuit is essential for generating rhythmic activity and the variability of motor programs while in the same time preventing temporal overlap of the different activity patterns.

The logic for model building is sound and the simplified network is able to reproduce the basic feature of motor activity. As such, the model provides the basis upon which more complete models could be build, like those incorporating proprioceptive and exteroceptive inputs and that could reproduce other types of motor programs of other types of sequence of motor activity, as they describe in the discussion. Such models could generate testable predictions about how network activity generates motor programs and specifically transitions between motor programs in different context and could be of interest to a wide range of neuroscientists and specifically those studying motor systems, both theoreticians and experimentalists.

I onlly have minor suggestions for the authors:

The paper is overall well written and clear. Some sections of the results are sometimes overly descriptive. Streamlining those would, I believe, make the paper easier to read. Details could be kept in figure legends and figures.

For example : l199-l208, l525-l548, ...

Line 802 -hunching, should be defined (i.e. head retraction, a defensive actions…)

In some figure panels, the font is too small to read (e.g. Fig 1 A, Fig. 3A, Fig. 7A, Fig.8C)

Reviewer #3: The manuscript by Francis et al. describes the involvement of inhibitory circuit motifs in the diversity and variability observed in locomotion in fruit fly larvae. The authors conducted calcium imaging of glutamatergic neurons in the isolated central nervous system and analyzed the statistics and transitions of motor programs that appeared spontaneously. They observed that a posterior burst resets the generation rhythm of forward waves. The authors examined the motor programs generated in reduced preparations containing the A6 to A8 segments and found a diversity in motor programs. They found that rhythmic activity induced by a muscarinic agonist depends on the activity of GAD-GAL4 positive neurons. Based on these observations, the authors developed a computational model partially inspired by interneurons characterized previously in this system. Individual segments are modeled by one inhibitor unit (referring to GDL neurons) and two excitor units. One excitor (referring to A27h neurons) is for forward waves and the other is for backward waves (referring to A18b neurons). The model contains other components, such as feed-forward intersegmental excitation, command neurons for forward and backward wave initiation, initiation detector to avoid concurrent appearance of forward and backward waves, and wave detectors to prevent multiple waves at the same time. The authors observe several phenomena in the model mimicking larval motor patterns: Waves are interrupted by blocking excitors or activating inhibitors. When stochastic inputs are applied to command neurons, transitions of motor programs are observed. By incorporating thoracic circuits, the authors built a fully integrated circuit model. This model generates diversity and variability resembling activity patterns observed experimentally in the isolated CNS. The effect of distinct levels of disinhibition is similar between optogenetic experiments and the computer model. The authors show that the loss of function of different inhibitory motifs leads to distinct phenotypes in the computational model.

The physiological experiments and computer simulations are well-designed and properly executed. The theme that the authors are trying to address is quite important in neuroscience and could attract a wide range of researchers in both experiment and theoretical neuroscientists. However, there are several critical concerns regarding the manuscript that the authors need to clarify.

Major comments:

Page 15: GAD-Gal4 is used to express GtAcR1 in GABAergic neurons. However, the expression pattern of this driver is not cited or shown. Please clarify this.

Page 16: The authors state about hemisegments but the hemi-cord preparation is not used. This should be clarified.

Page 19: The hemisegmental inhibitory ("the inhibitor") is modeled referring to GDL. Although Fushiki et al. shows the necessity of GDL in forward waves, there is no evidence that shows GDL is required for backward waves. Some other evidence or observations supporting the existence of inhibitory interneurons modeled by "the inhibitor" should be described.

Page 20: The model includes feed-forward excitation based on Kohsaka et al. (2019). However, Kohsaka et al. (2019) doesn't report feed-forward motifs but feedback motifs. Biological evidence or observations supporting the intersegmental connection built in the model are required.

Page 24-27: The authors show that the model inspired by the Drosophila larval CNS combined with noise according to Poisson distribution exhibits diversity and variability. However, the involvement of inhibitory circuit motifs in diversity and variability in Drosophila larval locomotor system, as described in the title, is not shown. The data supporting the title statement is necessary.

Page 37: There are three issues on Figure 14: 1) The traces in Figure 14Ai look quite different from Figure 14Bi and 14Ci. Please explain this. 2) There is no effector control (CS>GtAcR1) to check the effect of the leaky expression of UAS-GtAcR1. 3) Quantitative analysis of the effect of blocking GABAergic neurons is missing. Readers cannot interpret how "representative" the data shown in Figure 14 are in the group of n = 4 preparations.

Page 41 and 42: The ability of the muscarinic receptor agonist to induce forward waves, which is reported in Jonaitis et al., 2022, is cited several times. However, how this finding is reflected in the model is not clear. Please describe this.

---

## [Editor Report · Decision Letter 2]

14 Jan 2025

Dear Stefan,

Thank you for your patience while we considered your revised manuscript "Generation of motor program diversity and variability through inhibitory circuit motifs in the Drosophila larval locomotor system" for publication as a Research Article at PLOS Biology. This revised version of your manuscript has been evaluated by the PLOS Biology editors and the Academic Editor.

Based on our Academic Editor's assessment of your revision, we are likely to accept this manuscript for publication, provided you satisfactorily address the following data and other policy-related requests:

* We would like to suggest a different title to improve its accessibility for our broad audience: Inhibitory circuit motifs in Drosophila larvae generate motor program diversity and variability

* Please add the links to the funding agencies in the Financial Disclosure statement in the manuscript details.

* Please note that per journal policy, we do not allow the mention of "data not shown", "personal communication", "manuscript in preparation" or other references to data that is not publicly available or contained within this manuscript. Please either remove mention of these data or provide figures presenting the results and the data underlying the figure (Figure 3).

* DATA POLICY:

Regardless of the method selected, please ensure that you provide the individual numerical values that underlie the summary data displayed in the following figure panels as they are essential for readers to assess your analysis and to reproduce it: 2EF, 3A, 4B, 5C, 6HI, 6II, 8C, 9B and S3B.

* CODE POLICY

We expect to receive your revised manuscript within two weeks.

*Published Peer Review History*

*Press*

Sincerely,

Christian

Christian Schnell, PhD

Senior Editor

cschnell@plos.org

PLOS Biology

---

## [Editor Report · Decision Letter 3]

14 Feb 2025

Dear Stefan,

Thank you for your patience while we considered your revised manuscript "Inhibitory circuit motifs in Drosophila larvae generate motor program diversity and variability" for publication as a Research Article at PLOS Biology.

I've just gone through our final editorial checks. Most things look alright, but there are three open points:

* I could not locate any of the data you mention in the Data Availability Statement. The repository at 10.17630/779141ce-c26a-483b-bfee-4f12cf71d7b2 does not seem to contain any data.

* Furthermore, I could not find the source data file. An example of how this can look like can be found in this recently published paper: https://journals.plos.org/plosbiology/article?id=10.1371/journal.pbio.3003002#sec044

Regardless of the method selected, please ensure that you provide the individual numerical values that underlie the summary data displayed in the following figure panels as they are essential for readers to assess your analysis and to reproduce it: 2EF, 3A, 4B, 5C, 6HI, 6II, 8C, 9B and S3B.

Please also ensure that figure legends in your manuscript include information on where the underlying data can be found, and ensure your supplemental data file/s has a legend, for example in the corresponding figure legends: "The data underlying these analyses are provided in S1 Data."

We expect to receive your revised manuscript within one week.

*Published Peer Review History*

*Press*

Sincerely,

Christian

Christian Schnell, PhD,

Senior Editor

cschnell@plos.org

PLOS Biology

---

## [Editor Report · Decision Letter 4]

3 Mar 2025

Dear Stefan,

Thank you for the submission of your revised Research Article "Inhibitory circuit motifs in Drosophila larvae generate motor program diversity and variability" for publication in PLOS Biology. On behalf of my colleagues and the Academic Editor, Bing Ye, I am pleased to say that we can in principle accept your manuscript for publication, provided you address any remaining formatting and reporting issues. These will be detailed in an email you should receive within 2-3 business days from our colleagues in the journal operations team; no action is required from you until then. Please note that we will not be able to formally accept your manuscript and schedule it for publication until you have completed any requested changes.

PRESS

Sincerely, 

Christian

Christian Schnell, PhD

Senior Editor

PLOS Biology

cschnell@plos.org